

**Assessment of lidar depolarization uncertainty by means of a polarimetric lidar simulator**

Bravo-Aranda, J.A.[1,2,3], Belegante L.[4], Freudenthaler V.[5], Alados-Arboledas A.[1,2], Nicolae D.[4], Granados-Muñoz M. J.[1,2], Guerrero-Rascado, J. L.[1,2], Amodeo A.[6], D'Amico G.[6], Engelmann R.[7], Pappalardo G.[6], Kokkalis P.[8], Mamouri R.[9], Papayannis A.[8], Navas-Guzmán F.[1,2,*], Olmo, F. J.[1,2], Wandinger U.[7] and Haeffelin, M.[3]

1 Andalusian Institute for Earth System Research (IISTA-CEAMA), Granada, Spain
2 Dpt. Applied Physics, University of Granada, Granada, Spain
3 Institute Pierre-Simon Laplace, CNRS-Ecole Polytechnique, Paris, France
4 National Institute of Research&Development for Optoelectronics - INOE 2000, Magurele, Ilfov, Romania
5 Ludwig-Maximilians-Universität Meteorologisches Institut, München, Germany
6 Consiglio Nazionale delle Ricerche Istituto di Metodologie per l'Analisi Ambientale I.M.A.A. - C.N.R., Potenza, Italy
7 Leibniz Institute for Tropospheric Research (TROPOS), Permoserstr. 15 04318 Leipzig, Germany
8 Laser Remote Sensing Unit, National Technical University of Athens, Physics Dept., 15780 Zografou, Greece
9 Department of Civil Engineering and Geomatics, Cyprus University Of Technology, Lemesos, Cyprus
* Institute of Applied Physics (IAP), University of Bern, Bern, Switzerland

*Correspondence to*: J. A. Bravo-Aranda (jabravo@ugr.es)

**Abstract.** Lidar depolarization measurements distinguish between spherical and non-spherical aerosol particles based on the change of the polarization state between the emitted and received signal. The particle shape information in combination with other aerosol optical properties allow the characterization of different aerosol types and the retrieval of aerosol particle microphysical properties. Regarding the microphysical inversions, lidar depolarization technique is becoming a key method since particle shape information can be used by algorithms based on spheres and spheroids, optimizing the retrieval procedure. Thus, the identification of the depolarization error sources and the quantification of their effects are crucial. This work presents a new tool to assess the lidar polarizing sensitivity and to estimate the systematic error of the volume linear depolarization ratio ($\delta$), combining the Stokes-Müller formalism and the Monte Carlo technique. This tool is applied to a synthetic lidar system and to several EARLINET lidars with depolarization capabilities at 355 or 532 nm. The results evidence that the lidar polarization sensitivity can lead to $\delta$ relative errors larger than 100%, being more probable its overestimation. The lidar systems show $\delta$ relative errors larger than 100% for $\delta$ values around the molecular one ($\sim$0.004), decreasing up to $\sim$10% for $\delta = 0.45$. However, among them, POLIS system shows the best behaviour with $\delta$ relative errors of 25% and 0.22% for $\delta = 0.004$ and $\delta = 0.45$, respectively, evidencing how a proper characterization of the lidar polarizing sensitivity can drastically reduce the $\delta$ systematic errors. In this regard, we provide some indications to reduce the lidar polarizing sensitivity and to improve its characterization.

## 1 Introduction

The lidar depolarization technique is a useful tool for different applications in atmospheric science such as the identification of the thermodynamic phase of clouds (e.g., Ansmann et al., 2005; Reichardt et al., 2003;



Schotland et al., 1971) and the aerosol typing (e.g., Bravo-Aranda et al., 2013; Gross et al., 2011, 2012, 2013, 2015; Navas-Guzmán et al., 2013). Additionally, lidar depolarization technique is very important for improving the retrieval of microphysical aerosol properties (e.g., Ansmann et al., 2011; Chaikovsky et al., 2002; Granados-Muñoz et al., 2014; Wagner et al., 2013, Samaras et al., 2015) becoming crucial for

inversion algorithms based on modelling aerosol particles as spherical and spheroids. Unfortunately, the reliability of the lidar depolarization technique is limited due to the complexity of the depolarization calibration. On one hand, relative depolarization calibration introduces a high uncertainty due to the low signal-to-noise ratio and thus, an absolute calibration is required. On the other hand, absolute calibration methods do not take into account all lidar polarizing effects, like the polarizing-dependent receiving

transmission (Mattis et al., 2009). In this sense, many authors have focussed their effort on the improvement of the lidar depolarization calibration (e.g., Álvarez et al., 2006; Bravo-Aranda et al., 2013; Hayman and Thayer, 2012) and on the determination of the depolarization uncertainties (e.g., Freudenthaler et al., 2009; Freudenthaler, 2016a). Therefore, the identification of the lidar polarizing sensitivity is relevant for i) an appropriate assessment of lidar depolarization results, ii) for a prioritizing of lidar Research & Development

and iii) for the development of future lidar generations. In this regard, Freudenthaler (2016a) provides a theoretical framework of the lidar depolarization technique based on the Stokes-Müller formalism including the formulae for the polarization calibration factor covering different calibration techniques and lidar setups.

This work quantifies the volume linear depolarization ratio uncertainty ($\Delta\delta$) due to the unknown systematic

errors caused by the lidar polarizing sensitivity and assesses the contribution of each lidar functional block to the total uncertainty. A software tool called *Polarimetric Lidar Simulator* (PLS) has been developed based on the theoretical framework presented by Freudenthaler (2016a) and the Monte Carlo technique. The PLS is applied to several lidar systems in order to show the dependence of the systematic error on their design features. Random errors due to signal noise are neglected in this work. Their contribution to the

uncertainty can be derived by means of Monte Carlo technique in a similar way to Pappalardo et al., 2004, Guerrero-Rascado et al., 2008.

In Sect. 2 we present the lidar performance description in terms of functional blocks and the approaches to the Stokes-Müller formalism. After that, in Sect. 3, the PLS performance is explained in detail. Then, the $\delta$ systematic error is analyzed according to two different approaches:

i)        in Sect.4, a synthetic lidar setup is used to quantify the corresponding $\Delta\delta$ and analyse the most important error sources;

ii)        in Sect.5, the $\Delta\delta$ is estimated for several EARLINET lidar systems and the error sources are analysed as well and to point to possible ways to reduce the uncertainties.

Finally, conclusions are reported in Sect. 6.



## 2. Basis of the Polarimetric Lidar Simulator (PLS): Stokes-Müller formalism applied to lidar functional blocks

As introduced by Freudenthaler (2016a), lidar systems can be subdivided in functional blocks: laser, laser emitting optics (beam expander, steering mirrors), receiving optics (telescope, collimator, dichroic mirrors…), and the polarizing beam splitter including the detectors. Furthermore, the depolarization calibrator can be considered as an additional functional block. Figure 1 shows a lidar scheme based on functional blocks including the laser beam ($I_L$), the emitting optics ($M_E$), the receiving optics ($M_O$), calibrator ($C$), rotator ($R$) and the polarizing splitter ($M_R$ and $M_T$) and the received signals ($I_R$ and $I_T$). This scheme is really useful for lidar modelling based on the Stokes-Müller formalism. Next sections explain each functional block, describe the assumptions and approaches performed by PLS and show their Stokes vectors or Müller matrices.

### 2.1 Laser, $I_L$:

Generally, lidar lasers produce linear polarized light being commonly assumed 100% linear polarized (i.e., polarizing parameter, $a_L = 1$). However, the purity of the polarization state of the laser light is usually unknown. Recently, it has been demonstrated that emitted laser beams could have elliptical polarization components affecting the depolarization measurements (Belegante, L. and Freudenthaler, V.; personal communication). Also, the misalignment angle of the laser polarizing plane with respect to the polarizing splitter incident plane ($\alpha$) can also affect the depolarization measurements since the rotated linear polarized beam become elliptical. Due to the lack of information about the possible range uncertainty of the elliptical polarization degree of the emitted laser beams, only $\alpha$ is analysed in this work being $I_L$ defined by

$$I_L = I_L \begin{pmatrix} 1 \\ cos(2\alpha) \\ sin(2\alpha) \\ 0 \end{pmatrix} \qquad (2.1)$$

where $I_L$ is the laser energy. This value is considered rather constant during typical accumulation time of lidar measurement and thus, it is fixed as $I_L = 1$, in all cases. Only short-term unstable laser intensity may affect depolarization measurements since, for example, laser intensity would change during the calibration process. However, Bravo-Aranda et al., 2013 show a very constant value of the depolarization calibration in a 6-month period which can be only reached with a good laser energy stability.

### 2.2 Laser emitting optics, $M_E$:

The emitting functional blocks, formed by a set of steering and dichroic mirrors, leads the laser beam to the atmosphere and, optionally, includes beam expanders for eye safety reasons and for increasing the illuminated portion of the atmosphere with respect to the field of view of the telescope. The polarizing effect of beam expanders was neglected since we can't estimate the uncertainties introduced by possible birefringence (as in the case of CaF2 lenses of apochromatic beam expanders). Additionally, windows in the transmitting part (roof window) are also neglected due to its complex analysis and lack of information. The effect of these optical devices has to be investigated in the future.

The general Müller matrix of steering and dichroic mirrors is





$$M = T \begin{pmatrix} 1 & D & 0 & 0 \\ D & 1 & 0 & 0 \\ 0 & 0 & Zcos(\Delta) & Zsin(\Delta) \\ 0 & 0 & -Zsin(\Delta) & Zcos(\Delta) \end{pmatrix} \qquad (2.2)$$

where $T$ and $D$ are the transmittance and the diattenuation defined by

$$T = \frac{T^p + T^s}{2} \qquad (2.3)$$

$$D = \frac{T^p - T^s}{T^p + T^s} \qquad (2.4)$$

with $T^p$ and $T^s$ as the parallel and perpendicular intensity transmission coefficients (transmittances) respect to the incident plane. The phase shift between parallel and the perpendicular components (hereafter, called phase shift) is noted by $\Delta$ and $Z$ is given by

$$Z = \sqrt{1 - D^2}, \qquad (2.5)$$

Assuming an emitting optics formed by different steering/dichroic mirrors, the Müller matrix of the
emitting optics, $M_E$, is expressed

$$M_E = M_1 M_2 \dots M_n = \prod_i M_i \qquad (2.6)$$

where the subscript $i = \{1,2, \dots n\}$ indicates number of steering/dichroic mirrors. From Eq. 2.6, it can be obtained, by matricial multiplication, an effective Müller matrix, $M_E$, as follow

$$M_E = T_E \begin{pmatrix} 1 & D_E & 0 & 0 \\ D_E & 1 & 0 & 0 \\ 0 & 0 & Z_E cos(\Delta_E) & Z_E sin(\Delta_E) \\ 0 & 0 & -Z_E sin(\Delta_E) & Z_E cos(\Delta_E) \end{pmatrix} \qquad (2.12)$$

where $T_E$, $D_E$ and $\Delta_E$ are the effective transmittance, diattenuation and phase shift of $M_E$. For example, considering an emitting functional block made by two dichroic mirrors ($M_1$ and $M_2$), $M_E$ is given by

$$M_E = T_{12} \begin{pmatrix} 1 & D_{12} & 0 & 0 \\ D_{12} & 1 & 0 & 0 \\ 0 & 0 & Z_{12} cos(\Delta_{12}) & Z sin(\Delta_{12}) \\ 0 & 0 & -Z_{12} sin(\Delta_{12}) & Z cos(\Delta_{12}) \end{pmatrix} \qquad (2.7)$$

where $T_{12}$, $D_{12}$, $Z_{12}$ and $\Delta_{12}$ are

$$T_{12} = T_1 T_2 (1 + D_1 D_2) \qquad (2.8)$$

$$D_{12} = \frac{D_1 + D_2}{1 + D_1 D_2} \qquad (2.9)$$

$$Z_{12} = \frac{Z_1 Z_2}{1 + D_1 D_2} \qquad (2.10)$$

$$\Delta_{12} = \Delta_1 + \Delta_2 \qquad (2.11)$$

with $T$, $D$, $Z$ and $\Delta$ subscripted by 1 and 2 are the parameters of $M_1$ and $M_2$ according to the Eq. 2.2. This process can be applied iteratively to obtain the effective Müller matrix, $M_E$ corresponding to emission block
composed by more than two steering or dichroic mirrors.





Since functional blocks are generally built as robust pieces, we assume the absence of rotational misalignments between the steering/dichroic mirrors. However, we consider a rotational misalignment of the whole functional block with respect to the polarizing beam splitter (PBS) incident plane. To this aim, the rotation Müller matrix, $\boldsymbol{R}$, is defined by

$$\boldsymbol{R}(\beta) = \begin{pmatrix} 1 & 0 & 0 & 0 \\ 0 & cos(2\beta) & -sin(2\beta) & 0 \\ 0 & sin(2\beta) & cos(2\beta) & 0 \\ 0 & 0 & 0 & 1 \end{pmatrix} \tag{2.13}$$

where the angle $\beta$ describes the rotational misalignment of $\boldsymbol{M}_E$ with respect to the PBS incident plane. Thus,

$$\boldsymbol{M}_E(\beta) = \boldsymbol{R}(\beta)\boldsymbol{M}_E\boldsymbol{R}(-\beta) \tag{2.14}$$

Resulting,

$$\boldsymbol{M}_E(\beta) = T_E \begin{pmatrix} 1 & D_E cos(2\beta) & D_E sin(2\beta) & 0 \\ D_E cos(2\beta) & \left(1 - W_E sin^2(2\beta)\right) & W_E sin(2\beta)cos(2\beta) & -Z_E sin(\Delta_E)sin(2\beta) \\ D_E sin(2\beta) & W_E sin(2\beta)cos(2\beta) & \left(1 - W_E cos^2(2\beta)\right) & Z_E sin(\Delta_E)cos(2\beta) \\ 0 & Z_E sin(\Delta_E)sin(2\beta) & -Z_E sin(\Delta_E)cos(2\beta) & Z_E cos(\Delta_E) \end{pmatrix} \tag{2.15}$$

with $W_E = 1 - Z_E cos(\Delta_E)$.

The polarization effect of $\boldsymbol{M}_E$ is described by the effective diattenuation ($D_E$), the effective phase shift ($\Delta_E$) and the rotational misalignment of the whole functional block with respect to the PBL polarizing plane ($\beta$).

### 2.3 Receiving optics, $\boldsymbol{M}_O$:

This functional block, formed by the telescope and steering/dichroic mirrors, leads the received signal to the photomultipliers and, in case of multiwavelength lidar, separate the received signal by wavelength. In the same way of the emitting optics, $\boldsymbol{M}_O$ can be described by a unique effective Müller matrix as it follows

$$\boldsymbol{M}_O(\gamma) = T_O \begin{pmatrix} 1 & D_O cos(2\gamma) & D_O sin(2\gamma) & 0 \\ D_O cos(2\gamma) & \left(1 - W_O sin^2(2\gamma)\right) & W_O sin(2\gamma)cos(2\gamma) & -Z_O sin(\Delta_O)sin(2\gamma) \\ D_O sin(2\gamma) & W_O sin(2\beta)cos(2\gamma) & \left(1 - W_O cos^2(2\gamma)\right) & Z_O sin(\Delta_O)cos(2\gamma) \\ 0 & Z_O sin(\Delta_O)sin(2\gamma) & -Z_O sin(\Delta_O)cos(2\gamma) & Z_O cos(\Delta_O) \end{pmatrix}, \tag{2.16}$$

where $T_O$, $D_O$ and $\Delta_O$ are the effective transmittance, diattenuation and phase shift of $\boldsymbol{M}_O$, respectively, and $\gamma$ describes the $\boldsymbol{M}_O$ rotational misalignment with respect to the PBS incident plane. The telescope polarization effects with small incidence angles of the light beam are neglected in this work (e.g., Clark and Breckinridge, 2011, Di et al., 2015). This approximation is valid for Cassegrain telescopes but for Newtonian ones (as in the case of the PollyXT lidars, see Engelmann et al., 2015). Furthermore, windows commonly used to protect the telescope may also affect the polarization but, it was not considered in the simulator since its effect is very difficult to evaluate due to the lack of information about their properties and the time-dependent behiour.





### 2.4 Polarizing beam splitters ($M_R$ and $M_T$):

The lidar depolarization technique involves the measurements of different polarizing components of the received signals. Typically the lidars with depolarization capabilities are based on linear polarizing lasers. As a consequence, the separation of polarizing components is commonly performed into parallel/perpendicular or total/perpendicular with respect to the polarizing plane of the emitting laser beam. In case of circular polarized lasers, the polarised components are clockwise/counter-clockwise. A polarizing splitter separates the received signal into reflected and transmitted signals depending on the specific polarizing components. Consequently, two Müller matrices are required to describe the reflection ($M_R$) and transmission ($M_T$) processes. Due to similarities in the shape of the matrices, the notation is simplified with the subscript $S = \{R, T\}$ with $M_S$ as it follows

$$M_S = T_S \begin{pmatrix} 1 & D_S & 0 & 0 \\ D_S & 1 & 0 & 0 \\ 0 & 0 & Z_S cos(\Delta_S) & Z_S sin(\Delta_S) \\ 0 & 0 & -Z_S sin(\Delta_S) & Z_S cos(\Delta_S) \end{pmatrix} \tag{2.17}$$

where $\Delta_S$ is the phase shift of $M_S, T_S, D_S$ are defined by,

$$T_S = \frac{S_p + S_s}{2} \tag{2.18}$$

$$D_S = \frac{S_p - S_s}{S_p + S_s} \tag{2.19}$$

with $S_p$ and $S_s$ as the parallel ($p$) and perpendicular ($s$) transmittance/reflectance, respectively, and $Z_S = \sqrt{1 - D_S}$.

Polarizing beam-splitter cubes, PBS, are commonly used to split the received lidar signal into polarizing components. For this type of splitter, it can be derived that,

$$T_p + R_p = 1 \tag{2.20}$$

$$T_s + R_s = 1 \tag{2.21}$$

Ideally, the PBS split light into two orthogonally polarized beams: parallel and perpendicular with respect to the PBL incident plane (i.e., $R_p = T_s = 0$ ). However, commercial PBS are not ideal optical devices and always transmit a fraction of perpendicular polarization component and reflect part of parallel polarization component. This phenomenon is called cross-talk as it has been studied previously (e.g., Álvarez et al., 2006; Freudenthaler et al., 2009; Snels et al., 2009). As it is shown in Freudenthaler (2016a), it is relatively easy reduce the cross-talk adding polarizer sheet filters after the PBS. The Müller matrix of the cleaned PBS, $M_S^{\#}$ is

$$M_S^{\#} = T_S^{\#} \begin{pmatrix} 1 & D_S^{\#} & 0 & 0 \\ D_S^{\#} & 1 & 0 & 0 \\ 0 & 0 & 0 & 0 \\ 0 & 0 & 0 & 0 \end{pmatrix} \tag{2.22}$$

where $D_S^{\#} = \{D_T^{\#} = 1, D_R^{\#} = -1\}$ and the superscript '#' indicates that the PBS is 'cleaned'.



Generally, no optical devices with diattenuation or retardance are present between the splitter and the PMT's and thus, the phase shift of the $M_S$ does not affect to the measurements since the signals registered by the photomultipliers correspond to the first element of the Stokes vector ($I_R$, $I_T$). Only if a not-well aligned linear polarizer sheet is placed behind the PBS, to 'clean' the cross-talk, $\Delta_S$ may affect. However,

even in this case, the $\Delta_S$ effect can be neglected for misalignment angles below 10° between the polarizer and the PBS. Therefore, $\Delta_S$ is not considered in this work.

**2.5 Rotator, $R$:**

The parallel and perpendicular polarizing components with respect to the polarizing plane of the emitting laser beam can be either the transmitted or the reflected signal depending on the axial rotation angle, $\Psi$,

between the polarizing plane of the incident light and the polarizing splitter incident plane. For $\Psi = 90°$, the reflected and transmitted signals corresponds to the parallel and perpendicular polarized components and vice versa for $\Psi = 0°$. In order to consider the $\Psi$ influence, a rotator, $R$, (Eq. 2.13) is included as a lidar functional block (Fig. 1).

**2.6 Photomultipliers, $\eta_R$, $\eta_T$:**

The reflected and transmitted signals are detected by the photomultipliers which perform the light-to-electrical signal conversion. They affect the depolarization measurements as, in general, different photomultipliers have different gains. Regarding the Stokes-Müller formalism, the photomultiplier gains of the transmitted and reflected signals are modelled by the scalar values, $\eta_R$ and $\eta_T$. These gains tend to be rather stable as shown by Bravo-Aranda et al., (2013), providing very stable calibration factors over 6

20    months, therefore, their influence can be neglected being their values set at 1.

**2.7 Calibrator, $C$:**

The calibrator allows the determination of the polarizing effect of those optical devices located behind the calibrator and the differences between the PMT's gains. According to Fig. 1, the calibrator factor, $\eta$, includes the effects of the polarizing splitter ($M_R$ and $M_T$) and photomultiplier gains, $\eta_R$ and $\eta_T$

$$\eta = \frac{\eta_R}{\eta_T}\frac{T_R}{T_T},$$    (2.23)

Different calibration methods have been proposed either using the theoretical value of molecular depolarization (Cairo et al., 1999) or using additional optical devices like half-wave plates or polarization filters (e.g., Álvarez et al., 2006; Snels et al., 2009; Freudenthaler et al., 2009).

Particularly, the $\Delta90°$-calibration method has been extensively implemented within EARLINET (e.g,

Freudenthaler et al. 2009; Nemuc et al., 2013; Mamouri and Ansmann, 2014; Bravo-Aranda et al., 2015). This method uses two measurements rotating the polarizing plane of the received signal at angles $\phi_1$ and $\phi_2$ around the nominal axial rotation ($\Psi$) with the constraint $|\phi_2 - \phi_1| = 90°$ (e.g., $\phi_1 = 45°$, $\phi_2 = -45°$ around the measurement position $\Psi = 0°$). These rotations allow the equalization of the reflected and transmitted signals and thus, any difference between the reflected and transmitted signals is due only to the

polarizing effects of the optical devices between the calibrator and the photomultipliers. The equalization





of the reflected and transmitted signals can be made by a physical rotation of the receiving optics including the photomultipliers, by rotating an half wave plate placed before the PBS or by using a linear polarizing filter rotated accordingly.

The measured calibration factor, $\eta^*$, is calculated by

$$\eta^*(\Psi, x45° + \varepsilon) = \frac{I_R(\Psi, x45° + \varepsilon)}{I_T(\Psi, x45° + \varepsilon)} \tag{2.24}$$

where the two rotation angles, $\phi_1$ and $\phi_2$ are written as $x45° + \varepsilon$ with $x = \pm1$ indicating the rotational direction and $\varepsilon$ takes into account the error in determining the rotational angles. Then, their geometric mean

$$\eta^*_{\sqrt{\mp}}(\Psi, \varepsilon) = \sqrt{\eta^*(\Psi, +45° + \varepsilon)\eta^*(\Psi, -45° + \varepsilon)} \tag{2.25}$$

is calculated since it is less influenced by $\varepsilon$ than $\eta^*(\Psi, x45° + \varepsilon)$ as indicated by Freudenthaler et al. (2009).

It is possible to show (Freudenthaler, 2016a) that the analytical expressions of $\eta^*_{\sqrt{\mp}}(\Psi, \varepsilon)$ corresponding to different experimental setup are always in the form

$$\eta^*_{\sqrt{\mp}}(\alpha, D_E, \Delta_E, \beta, D_0, \Delta_0, \gamma, \varepsilon_r, D_T, D_R) = \eta f(\alpha, D_E, \Delta_E, \beta, D_0, \Delta_0, \gamma, \varepsilon_r, D_T, D_R) \tag{2.26}$$

where $f(\alpha, D_E, \Delta_E, \beta, D_0, \Delta_0, \gamma, \varepsilon_r, D_T, D_R)$ is the lidar polarizing sensitivity not corrected by $\eta$. Hereafter, the correction factor, $f(\alpha, D_E, \Delta_E, \beta, D_0, \Delta_0, \gamma, \varepsilon_r, D_T, D_R)$ will be noted by $f(\alpha, \dots)$.

**2.8 Reflected and transmitted signals, $I_R$, $I_T$:**

According to the Stokes-Müller formalism the reflected ($I_R$) and transmitted ($I_T$) signals can be obtained by multiplying the laser beam Stokes vector ($I_L$) by the subsequent Müller matrices which represents the different functional block and the atmosphere, $F$, which is described by

$$\boldsymbol{F} = F_{11}\begin{pmatrix} 1 & 0 & 0 & 0 \\ 0 & a & 0 & 0 \\ 0 & 0 & -a & 0 \\ 0 & 0 & 0 & (1-2a) \end{pmatrix} \tag{2.27}$$

where $F_{11}$ is the backscatter coefficient and $a$ is the polarization parameter. Despite both parameters are range-dependent, this dependence is omitted for the sake of clarity. Therefore, the lidar and calibration measurement Stokes vector are described by,

$$\boldsymbol{I}_S(\Psi) = \eta_S \boldsymbol{M}_S \boldsymbol{R}(\Psi)\boldsymbol{M}_o \boldsymbol{F}\boldsymbol{M}_E \boldsymbol{I}_L \tag{2.28}$$

$$\boldsymbol{I}_S(\Psi, x45° + \varepsilon) = \eta_S \boldsymbol{M}_S \boldsymbol{R}(\Psi)\boldsymbol{C}(x45° + \varepsilon)\boldsymbol{M}_o \boldsymbol{F}\boldsymbol{M}_E \boldsymbol{I}_L \tag{2.29}$$

where the first element of the Stokes vectors is the energy detected by the photomultipliers. Based on the Stokes-Müller formalism presented, the detected energy value depends on the 18 lidar parameters summarized in Table 1. However, only some of them are considered by PLS for the lidar polarizing sensitivity (see 'error source' column in Table 1).



**2.9 Volume linear depolarization ratio, $\delta$:**

The polarization parameter, presented in the previous section, is directly related to the volume linear depolarization ratio, $\delta$, by,

$$\delta = \frac{1-a}{1+a} \tag{2.30}$$

Both parameters informs about the particle shape. Specifically, larger $\delta$ values indicate less particles sphericity. For further details regarding mathematical expressions of depolarization parameters, see Cairo et al. (1999). $\delta$ can retrieved from lidar measurements by the following general equation to given by Freudenthalter, (2016a),

$$\delta = \frac{\delta^*(G_T+H_T)-(G_R+H_R)}{(G_R-H_R)-\delta^*(G_T-H_T)} \tag{2.31}$$

where the parameters $G_T$, $G_R$, $H_T$ and $H_R$, are determined solving the matrix multiplication of Equation 2.24 and separating the energy measured, $I_S$, by the polarization parameter, $a$, as follow

$$I_S = G_S + aH_S \tag{2.32}$$

and $\delta^*(\Psi)$ is the reflected-to-transmitted signal ratio divided by the calibration factor, $\eta$ (Eq. 2.23)

$$\delta^*(\Psi) = \frac{1}{\eta}\frac{I_R(\Psi)}{I_T(\Psi)} \tag{2.33}$$

where $\eta$ has to be derived from measured calibration factor, $\eta^*_{\sqrt{\mp}}$ (Eq. 2.26), estimating $f(\alpha,\dots)$ either from the lidar polarizing sensitivity information already available (e.g., from technical specifications), either from additional measurements performed to this aim (see Belegante et al., 2015) or assuming $f(\alpha,\dots)\sim1$ if there is not any characterization of the lidar polarizing sensitivity. Therefore, a better characterization of the lidar polarization sensitivity (through $G_S$, $H_S$ and $f(\alpha,\dots)$) leads to decrease the systematic errors on 20  lidar depolarization measurements.

**3 Polarimetric Lidar Simulator (PLS) performance**

In order to assess the lidar polarizing sensitivity and quantify the $\delta$ systematic error, the *Polarimetric Lidar Simulator (PLS)* has been developed based on the matrix equations resulting from the theoretical framework given by Freudenthaler (2016a) (see, Sect. 2). We use Monte Carlo technique to estimate $\delta$ since the 25  parameters involved in the simulation are not always independent. The PLS workflow is shown in Fig. 2 and explained in detail below.

1)  Creation of a parametric model: lidar parameters noted by $x_1,\dots,x_n$ in Fig 2 and listed in Table 1 are determined either from technical specifications of optical devices, or from manufacturer or assumed in the other case. Particularly, for the splitter properties, reflectance and transmittance coefficients 30  requires additional calculations according to the splitter type. MULHACEN, RALI, LB21 have commercial PBS and thus, $T_p$ and $T_s$ values and uncertainties from the technical specifications are used whereas the values and uncertainties of $R_p$, $R_s$ are calculated using the equation Eq. 2.20 and 2.21; MUSA, IPRAL and POLIS (355 and 532 nm) have a cleaned PBS so a suitable splitter is assumed (real values of $T_p$ and $R_s$ are only available for POLIS); for POLLY-XT SEA: $T_p$ and $T_s$ values and uncertainties are obtained from the



technical specifications, $R_p$ is calculated by the means of the extinction ratio ($ER$) as, $R_p = ER(1 - T_p)$, being ER the extinction ratio of the linear polarizer used to measure the perpendicular signal and $R_s = 0$ is assumed due to the high quality of the linear polarizer. Finally, the radiation-atmosphere interaction is simulated by the atmospheric parameters, $F_{11}$ (backscattering coefficient) and $\delta_r$ (real atmospheric $\delta$) based on the Stokes-Müller formalism (see Section 2.8).

2)      Calculation of the correction factors, $G_R$, $G_T$, $H_R$, $H_T$, and $f(x_1, , ...)$ based on the parameters values $x_1, ..., x_n$.

3)      Generation of simulated values: Gaussian or uniform distributions with a large number of random values (~100 or larger) are commonly used in order to obtain a reproducible $\Delta\delta$. In our case, lidar parameter uncertainties are not related to random variations but to a lack of knowledge of the true value and thus, uniform distribution is used since all the combinations are equality probable. However, a large number of simulation for each lidar parameters would lead to an unmanageable quantity of combinations ($10^{100}$) and thus, we adjust the number of iteration to the impact of each lidar parameter on $\delta$ as it follows

   a.   For the parameters $\alpha$, $D_E$, $\beta$, $\gamma$, and $\varepsilon$, we use only three values: $x_{i,j} = [x_i - \Delta x_i, \ x_i, \ x_i + \Delta x_i]$.

   b.   For the parameters $\Delta_E$, $D_0$, $\Delta_0$, $D_T$, and $D_R$, we use values between $x_i - \Delta x_i$ and $x_i + \Delta x_i$ which a fixed step calculated to provide around $10^6$ combinations: $x_{i,j} = [x_i - \Delta x_i, \ ..., x_i, \ ..., x_i + \Delta x_i]$.

4)      Evaluation of the model for each $x_{i,j}$ combination and the atmospheric parameters, $F_{11}$ and $\delta_r$. In order to estimate the $\delta$ influence on its systematic error, this procedure is performed at 0.004 and 0.45 as representative values of the minimum and maximum atmospheric $\delta$ values:

   a.   Calibration and measurement signals $I_s^{i,j}(\Psi)$ and $I_s^{i,j}(\Psi, \phi)$.

   b.   Calibration signals are used to retrieve the calibration factor, $\eta^{*,j}_{\sqrt{\pm}}$.

   c.   $\delta_j^*$ is retrieved using $\eta^{*,j}_{\sqrt{\pm}}$ and $f(\alpha, , ...)$.

   d.   $\delta_{s,j}$ is retrieved by means of Eq. 2.31.

5)      Coloured squares highlight the workflow linked to the calibration (red) and to the correction performed thanks to the characterization of the lidar polarizing sensitivity (green).

6)      The analysis of the results is performed in three different ways:

   a.   The uncertainty propagation of each simulation parameter, $x_i$, is analysed through the simulated-to-real $\delta$ difference, $E_\delta(x_i) = \delta_{s,j} - \delta_r$, varying $x_i$ within its uncertainty range $[x_i - \Delta x_i, x_i + \Delta x_i]$ while all the other parameters are kept. This method is used in Sect. 4.

   b.   Monte Carlo technique is commonly used to determine the uncertainty range by means of the standard deviation of the solution set. This can be performed when the number of combination is large enough and when the error source is random. In our case, this approximation would lead to erroneous $\Delta\delta$ since the lidar parameters uncertainties are not related to random variations but to a lack of knowledge of the true value. Therefore, for the sake of robustness, we determine the



depolarization uncertainty, $\Delta\delta$, as the minimum and maximum of the $\delta$ simulation set, $\left[\min\left(\delta_s^{i,j}\right), max\left(\delta_s^{i,j}\right)\right]$.

c. Finally, we analyse the simulated-$\delta$ frequency distribution by means of histograms, $histogram\left(\delta_s^{i,j}\right)$, where simulated-$\delta$ displacement toward larger (smaller) than the $\delta$ reference indicates overestimation(underestimation) of $\delta$.

**4 Depolarization uncertainties according to synthetic functional blocks**

In this section, we simulate a synthetic lidar system in order to evaluate the $\Delta\delta$ caused by each functional block. We choose the $\Delta90°$-calibration method implemented by means of a physical rotator in front of the PBS. The synthetic lidar system is based on different technical specifications of commercial optical devices (Table 2). It is worthy to note the large uncertainty of the effective phase shift ($\pm180°$) due to the knowledge about this property is in general poor.

The uncertainty propagation of each simulation parameter, $x_i$, is analysed through the simulated-to-real $\delta'$ difference $\left(E_\delta(x_i) = \delta_{s,j} - \delta_r\right)$ ranging $x_i$ within its uncertainty range $[x_i - \Delta x_i, x_i + \Delta x_i]$ while all the other parameters are kept. Then, $E_\delta(x_i)$ is parameterized using different $\delta_r$ values to analyze the atmospheric depolarization influence. To analyze the relationship between two different parameters, $E_\delta(x_i)$ is also parameterized with different values of other simulation parameter. Finally, the Monte Carlo technique is used to estimate $\Delta\delta$ from the set of $\delta$ solutions.

**4.1 Synthetic lidar: laser functional block analysis**

The laser may introduce errors in the depolarization measurements due to a misalignment angle of the laser polarizing plane with respect to the PBS incident plane ($\alpha$) (see Eq. 2.1). Figure 2 shows $E_\delta$ due to $\alpha$, $E_\delta(\alpha)$, parameterizing different values of $\delta$. As it can be seen in Fig. 3, $E_\delta(\alpha)$ increases with $\alpha$ in absolute terms. $\delta$ systematic error caused by $\alpha$, $\Delta\delta(\alpha)$, can be figure out by the minimum and the maximum of $E_\delta(\alpha)$. $\Delta\delta(\alpha)$ ranges between [0, 0.031] and [0, 0.024] for $\delta$ values of 0.004 and 0.45, respectively, showing a low $\delta$ dependence. Since $\delta_m$ is of the order of $10^{-3}$, we assume that $\Delta\delta(\alpha)$ can be neglected ($<1\cdot10^{-4}$) if $\alpha$ is fixed in the range $0° \pm 0.6°$.

**4.2 Synthetic lidar: emitting functional block analysis**

$\boldsymbol{M}_E$ is characterized by the effective diattenuation, $D_E$, and phase shift, $\Delta_E$, as well as the angle, $\beta$, describing the rotational misalignment of $\boldsymbol{M}_E$ with respect to the PBS incident plane. These parameters are dependent among them and thus, Fig. 4(top and bottom) shows $E_\delta$ dependence on $\Delta_E$ and $D_E$ parameterizing $\beta$. Additionally, the influence of the atmospheric depolarization is also assessed throughout $\delta$ values: 0.004 and 0.45. As it was aforementioned, $\Delta_E$ varies in the range [-180°, 180°] because the phase shift of steering and dichroic mirrors is generally not provided in the majority of the technical specifications.

As it can be seen in Fig. 4 (top) indicates that $D_E$ also introduces systematic errors except for $\beta = 0°$. According to this figure, $\Delta\delta(D_E, \beta) = [0, 0.001]$ showing a considerably decreases with $D_E$. Figure 4





(bottom) shows that $E_\delta(\Delta_E, \beta)$ is larger than 0.03 for $\beta$ larger than 5° in absolute terms indicating that the lack of information of $\Delta_E$ can lead to huge uncertainties even larger than 0.1 for $\beta$ larger than 10°.

Figure 4 shows also the influence of $\delta$ on $E_\delta(D_E, \beta)$ and $E_\delta(\Delta_E, \beta)$. As larger $\delta$ values decreases $\Delta\delta$, the use of laser emitting optics is less recommendable for studying aerosol types with low depolarization capabilities. In summary, the total $\Delta\delta$ due to the $\boldsymbol{M}_E$ polarizing effects is $[0, 0.13]$ for $\delta = 0.004$ and $[0, 0.1]$ for $= 0.45$. In order to avoid this error source, it is highly recommended do not use laser emitting optics if possible. If used, it is crucial to set $\beta = 0°$ to keep $\Delta\delta(\boldsymbol{M}_E)$ as lows as possible independently of the effective diattenuation and phase shift. Fort this example, $\beta = 0° \pm 2.5º$ would lead to neglect the $E_\delta(D_E, \beta)$ and $E_\delta(\Delta_E, \beta)$ (i.e., lower values than $1 \cdot 10^{-4}$).

### 4.3 Synthetic lidar: receiving functional block analysis

The parameters of the receiving optics ($\boldsymbol{M}_o$) are the effective diattenuation $D_o$ and phase shift $\Delta_o$, and the misalignment angle between the receiving optics and the PBS incident plane, $\gamma$. As in the case of $\boldsymbol{M}_E$ ($\beta$, $D_E$, $\Delta_E$), the influence of any of these parameters on $E_\delta$ is not independent. However, the relationship between $D_o$ and $\gamma$ is very weak and thus, Fig. 5(top) depicts $E_\delta$ versus $D_o$ parameterized by $\delta$ whereas Fig. 5(bottom) depicts $E_\delta$ versus $\Delta_o$ parameterized by $\gamma$. Both figures include the atmospheric depolarization by means of two $\delta$ values: 0.004 and 0.45.

According to Fig. 5 (top), $E_\delta(D_o)$ considerably increases with $D_o$ reaching values around 0.09 for $D_o$ around 0.1. Additional simulations (not shown) revealed that $E_\delta(D_o)$ is extremely large for $D_o$ larger than 0.15 in absolute terms highlighting the large impact of the receiving optics diattenuation on the depolarization measurements. Also, $E_\delta(D_o)$ increases with $\delta$ and thus, it is especially important for atmospheric aerosol with high depolarization (e.g., mineral dust or volcanic ash).

Negative values of $D_o$ causes larger $E_\delta(D_o)$, in absolute terms, than positive ones (e.g., $|E_\delta(D_o = -0.2)| = 0.17$ whereas $|E_\delta(D_o = +0.2)| = 0.11$ considering $\delta = 0.25$ in both cases) because the parallel signal is stronger than the perpendicular one. In order to neglect the $\boldsymbol{M}_o$ effect, $D_o$ uncertainty should be lower than $\pm 0.0010$ (i.e., $E_\delta(D_o) < 10^{-4}$) requiring of an extreme high precision. Thus, we advise the use of calibration methods which correct for $D_o$ or the experimental determination of this value as indicated by Belegante et al., (2015).

Figure 5 (bottom) shows the similarities between $E_\delta(\Delta_o)$ and $E_\delta(\Delta_E)$ with $\Delta\delta'(\Delta_o)$ larger than 0.03 for $\gamma = \pm 5°$. Therefore, it can be concluded that it is highly recommended to fix $\gamma = 0°$.

Summarising, $\Delta\delta(\boldsymbol{M}_o)$ would be $[-0.07, 0.12]$ and thus, it is very important to carefully determine the parameters $\gamma$ and $D_o$ of the receiving optics.

### 4.4 Synthetic lidar: polarizing splitter functional block analysis

For the synthetic lidar, we consider a non-cleaned polarizing beamsplitter which $T_T^p$, $T_R^p$, $T_T^s$ and $T_R^s$ values and uncertainties are shown in Table 2. Since $T_T^p + T_R^p = 1$ and $T_T^s + T_R^s = 1$, $T_T^p$, $T_R^p$, $T_T^s$ and $T_R^s$ uncertainties are not independent allowing the $E_\delta$ analysis using only $T_p$ parameterizing $T_s$. $\delta'$ values of 0.004 and 0.45 are used to study the influence of the atmospheric depolarization (Fig. 6). $E_\delta(T_T^p, T_T^s)$ has



a stronger dependence with the $T_T^s$ uncertainty, $\Delta T_T^s$, than with the $T_T^p$ uncertainty, $\Delta T_T^p$, being the ratio $E_\delta(\Delta T_T^s)/\Delta T_T^s$ around three times the ratio $E_\delta(\Delta T_T^p)/\Delta T_T^p$ because the parallel signal intensity is larger than the perpendicular one. Thus, the cross-talk effect is more critic on perpendicular signal than on the parallel one. $\Delta\delta(T_T^p, T_T^s)$ strongly increases with $\delta$. For $T_T^s = 0.01$ (dashed line, Fig. 6), $\Delta\delta(T_T^p)$ increases one order

of magnitude (from $\Delta\delta(T_T^p) = [-0.001, \ 0.0013]$ to $\Delta\delta(T_T^p) = [-0.013, \ 0.012]$) between $\delta = 0.004$ (thick lines) and $\delta = 0.45$ (thin lines). This contamination can be avoided by means of cleaned PBS (see Sect. 2.4) since this setup strongly reduces this uncertainty source.

**4.5 Synthetic lidar: calibrator functional block analysis**

The $\Delta 90°$-calibration method with a rotator located in front of the PBS is considered. This calibration
method is summarized in Sect.2.7 (see Freudenthaler (2016a) and Belegante et al., (2015) for further details). It is worthy to note that the uncertainties due to a physical rotational of part of the lidar system affects both calibration and normal measurements whereas, the use of a complementary optical device for the calibration (e.g., polarizing filter) only affects the calibration measurements decreasing the systematic error due to the rotation uncertainty.

As it can be seen in Fig. 7, $\Delta\delta(\varepsilon_r)$ is $[0, \ 0.0008]$ for $\delta = 0.004$ and $[0, \ 0.0006]$ for $\delta = 0.45$. Considering the large uncertainty range ($\varepsilon_r = \pm 5°$), $\Delta\delta(\varepsilon_r)$ is really low. Since $\varepsilon_r$ is used to be lower than 1° with uncertainties ranging between 0.1° and 3 arcminutes, it can be concluded that $\Delta\delta(\varepsilon_r)$ can be neglected in most cases. This happens because the $\Delta 90°$-calibration method is used and thus, for other calibration methods this parameter needs to be taken into account.

Despite the fact that $\varepsilon_r$ does not introduce large uncertainties, the position of calibrator within the lidar system has to be considered. For example, the $\Delta 90°$-calibration in front of the receiving optics corrects for the diattenuation of the receiving optics, $D_o$ but it does not if it is located behind it. Belegante et al., (2015) highlight this effect by comparison of the $D_o$ retrieved in several EARLINET lidars and show an experimental setup which allows the determination of $D_o$.

**4.5 Synthetic lidar: total uncertainty analysis**

The total $\delta$ systematic error ($\Delta\delta$), including all possible correlations, is estimated by using the Monte Carlo technique based on uniform distribution. In order to keep the number of iteration around $10^6$, $\alpha$, $D_E$, $\beta$, $\gamma$, and $\varepsilon$ are simulated with three values each one (e.g., $\alpha = [-10°, 0°, 10°]$) and $\Delta_E$, $D_0$, $\Delta_0$, $D_T$ and $D_R$ with five values each one (e.g., $\Delta_0 = [-180°, -90°, 0°, °, 90°, 180°]$). As a result, there are $3^5 5^5 \cong 7.6 \cdot 10^5$
simulated $\delta$ values. Figure 8 shows the $\delta$ histograms at $\delta = 0.004$ (left) and $\delta = 0.45$ (right) and the minima and maxima are provided in Table 3. For both simulations the values obtained for $\delta$ span over a quite large range reaching even unrealistic negative values or $\delta$ values larger than 1. As typical $\delta$ values are in the range $[0.04, \ 0.1]$ (e.g., Gross et al., 2011; Murayama et al., 2004) for biomass burning aerosol and in the range $[0.150, \ 0.3]$ for mineral dust (e.g., Gross et al., 2011), it can be concluded that the
hardware polarization sensitivity can affect the depolarization results causing relative errors even larger than 100%. Since the $\delta$ distribution is displaced to the right of the $\delta$ reference, the overestimation of $\delta$ is more probable than the underestimation.



### 5 Depolarization uncertainty assessment in the framework of EARLINET

PLS is applied to the eight EARLINET lidar systems listed in Table 4. Detailed information about the analysed lidar systems is given by Freudenthaler, (2015b) and Wandinger and al., (2015) except for IPRAL (IPSL high-Performance multi-wavelength RAman Lidar for Cloud Aerosol Water Vapor Research) which

have been recently deployed at SIRTA atmospheric research observatory (Haeffelin et al., 2005). IPRAL provides measurements at 355 (parallel and perpendicular polarizing components), 532 and 1064 nm (elastic backscatter) and at 387 (from $N_2$), 408 (from $H_2O$) and 607 nm (from $N_2$) (Raman-shifted backscatter).

Table 5 shows the values and uncertainties of the lidar parameters used for simulation. Main differences

among lidars are: i) the use of steering optics (e.g., MUSA and POLIS do not have this optional functional block), ii) the position and type of the calibrator (e.g., MULHACEN and RALI: polarizer in front of $\mathbf{M_o}$, and MUSA: waveplate in front of $\mathbf{M_S}$), iii) the $\mathbf{M_S}$ type and polarizing components (e.g., POLLY-XT SEA: polarizer providing the total and perpendicular polarizing components and MUSA: PBS providing the parallel and perpendicular ones), iv) the different values of certain parameters (e.g., $D_o = 0.35$ and $D_o =$

$-0.001$ in MULHACEN and POLIS 355nm, respectively) and v) the different uncertainties on polarization sensitivity parameters i.e., the phase shift $\Delta_o$ uncertainty is negligible for IPRAL system and much larger ($\pm 180°$) for all other lidars.  It is worthy to note that MULHACEN, RALI and LB21 have been already upgraded with a cleaned PBS (Belegante et al., 2015). However, in order to highlight the cross-talk effect, the present analysis is based on the previous lidar configuration.

The number of Monte Carlo iterations of each lidar property is calculated based on uniform distribution to provide around $10^6$ combinations. MULHACEN, RALI, LB21, and POLLY-XT SEA are simulated using three values for $\alpha$, $D_E$, $\beta$, $\gamma$, and $\varepsilon_r$, $[x_i - \Delta x_i,\ x_i,\ x_i + \Delta x_i]$, and using five values for $\Delta_E$, $D_0$, $\Delta_0$, $D_T$, and $D_R$, $[x_i - \Delta x_i,\ ...,x_i,\ ...,x_i + \Delta x_i]$ with a fixed step (see more details in Sect. 3), resulting a total number of combinations of $3^5 5^5 \sim 7.6 \cdot 10^5$. Since POLIS and MUSA do not have $\mathbf{M_E}$, the contribution of $D_T$ and

$D_R$ can be neglected. For these systems the simulation is run assuming three possible values for the parameters $\alpha$, $\gamma$, and $\varepsilon_r$ and 193 values for the parameters $D_0$ and $\Delta_0$ resulting in total number of combinations of $3^3 193^2 \sim 1 \cdot 10^6$. Finally for IPRAL, the uncertainty of $\Delta_0$, $D_T$ and $D_R$ is neglected, and thus, $\alpha$, $D_E$, $\beta$, $\gamma$, and $\varepsilon_r$ are simulated with three values whereas $D_0$ and $\Delta_E$ are simulated with 65 values. A total of $3^5 65^2 \sim 1 \cdot 10^6$ combinations are computed.

Figure 9 and 10 show the $\delta$ histograms at $\delta = 0.004$ and $\delta = 0.45$, respectively, for the EARLINET lidars (Table 4). The histogram shapes are quite different among the lidar systems. MULHACEN and RALI show a like-Gaussian distribution shape almost centred at $\delta$ reference whereas MUSA, IPRAL and POLLY-XT SEA at $\delta = 0.004$ and POLIS at at $\delta = 0.004$ and at $\delta = 0.045$ show an irregular one (POLIS-distribution figures with adapted axis are included as supplement). Moreover, LB21 and POLLY-XT SEA distributions

at $\delta = 0.45$ show discrete like-Gaussian shapes. To explain this behaviour, the role played by $D_o$ (for LB21 lidar and POLLY-XT SEA), $\alpha$ (for MULHACEN) and $\varepsilon_r$ (for RALI) in the total $\delta$ uncertainty is reported in Fig. 10. As it can be seen, $\delta$ sub-histogram due to $\varepsilon_r$ at RALI has the same shape of the $\delta$ histogram. However, $\delta$ sub-histograms due to $D_o$ (at POLLY-XT SEA and LB21) and $\alpha$ (MULHACEN) show a



displacement according to their values. Such analysis indicates that the discrete shape is due to the low iteration number of lidar parameters with a large impact on $\delta$. Therefore, the sub-histograms allow the identification of the more relevant lidar parameters for each lidar system. In these cases, $D_o$ uncertainty has a large impact on the $\delta$ distribution in POLLY-XT SEA and LB21 (leading to the aforementioned discrete

distribution); $\alpha$ uncertainty leads to a wider $\delta$ distribution for MULACHEN indicating a medium relevance; and $\varepsilon_r$ does not differ from the RALI's $\delta$ histogram, indicating that other parameters are more relevant (e.g., $D_T$ or $D_R$).

As aforementioned, lidar parameters uncertainties are not related to random variations but to a lack of knowledge of the true value. Therefore, we establish the depolarization uncertainty, $\Delta\delta$, as the minimum

and maximum of the simulated $\delta$ set since each combination of lidar parameters has the same probability to be the real one (see Table 6). For $\delta$ of the order of magnitude of the molecular volume linear depolarization ratio ($\sim$0.004), relative errors are larger than 100 % in all the lidars except for POLIS at 355 nm and 532 nm with $\Delta\delta/\delta \sim$25%. However, relative errors decreases for $\delta = 0.45$ being between 3 and 10% for all the lidars except for POLIS at 355 nm and 532 nm with $\Delta\delta/\delta \sim$0.16%. The large difference of

$\Delta\delta$ between POLIS and the other analysed lidars mainly relay on i) the special dichroic mirrors designed to have almost negligible diattenuation, ii) the absence of steering optics, iii) the cleaned PBS and iv) the low angle uncertainties (Table 5).

Despite $\Delta\delta$ is still large, mainly for low $\delta$, it is worthy to refer the improvement with respect to a lidar without any polarizing sensitivity characterization (e.g., synthetic lidar). This result support the effort

carried out by the EARLINET community towards a better characterization of lidar polarizing sensitivity. According to the results described in the previous sections, some indications can be provided to reduce the lidar polarizing sensitivity on lidar systems. For example, the laser beam ($I_L$) polarization purity could be improved by using a high-energy polarizing filter between the emission and the laser emitting optics. To reduce the uncertainty introduced by $M_E$, it is highly recommended to avoid this optional functional device

pointing the laser beam directly to the atmosphere. The $\Delta\delta$ due to $M_E$, and $M_o$ can be decreased by improving their rotational alignment (i.e., $\beta$ and $\gamma$) with respect to the polarizing splitter. Finally, the PBS cross-talk can be removed by using cleaned PBS. If these improvements cannot be performed, a good characterization of the lidar polarizing sensitivity can drastically reduce $\Delta\delta$. For example, the $\alpha$ and $D_0$ values can be determined by experimental assessments as indicated by Belegante et al., (2015).

## 6 Conclusions

This work analyses the lidar polarizing sensitivity by means of the Stokes-Müller formalism and provides a new tool to quantify the systematic error of the volume linear depolarization ration ($\delta$) using the Monte Carlo technique.

The synthetic lidar simulation showed that $\delta$ could range between [-0.025, 1.1064] and [0.386, 1.021] for $\delta$ values of 0.004 and 0.45, respectively. As typical atmospheric $\delta$ values range between 0.05 and 0.3, it can be concluded that the lidar polarization sensitivity affects the depolarization measurements strongly





leading to unrealistic $\delta$ values. $\delta$ histogram analysis showed that larger simulated $\delta$ values than the reference one are more frequent than the lower ones, and thus, it can be concluded that the lidar polarizing sensitivity usually overestimates $\delta$.

Thus, a proper characterization of each functional block is crucial for the lidar depolarization technique.
The most critical functional blocks are the receiving optics ($\boldsymbol{M}_O$) and the splitter $\boldsymbol{M}_S$ being the effective diattenuation of the receiving optics ($D_o$), the most important issue. Then, the emitting and receiving optics phase shifts and the rotational misalignment, between the polarizing plane of the laser and the incident plane of the PBS, are also relevant.

EARLINET lidar simulations show $\delta$ relative errors varying from >100% to ~10% for $\delta$ between the order
of magnitude of the molecular depolarization to $\delta = 0.45$ for most of the analysed lidars. Despite $\delta$ uncertainties are large for low $\delta$ values, it is worthy to note the effort performs by the EARLINET community to improve and characterize the lidar polarizing sensitivity of their lidar systems.

The uncertainty of some parameters (e.g., phase shift uncertainty of dichroic mirrors) is very large because, in general, the optical manufacturers do not provide specific information. Studies like the one presented in
the work, identify which parameters need more accurate characterization and may be the trigger to develop new lidar systems with better performance in depolarization measurements. An example is the receiving optics of IPRAL, the lidar installed at the SIRTA atmospheric research observatory, designed by RAYMETRICS (Athens, Greece) with the help of specialized companies, providing special dichroic mirrors with almost negligible diattenuation and 0° phase shift as a consequence of the lidar polarizing
sensitivity studies carried out in the framework of EARLINET.

Finally, further investigations are still required for a better understanding of the polarizing effects of windows, lenses and Newtonian telescopes. Furthermore, the elliptical polarization in the outgoing laser beam may strongly affect the $\delta$ determination. Experimental $\delta$ values out of the simulated $\delta$ distribution may be understood like an evidence of the effect of these optical devices.

**Acknowledgements**

This work was supported by the Andalusia Regional Government through projects P12-RNM-2409 and P10-RNM-6299, by the Spanish Ministry of Science and Technology through projects CGL2010-18782, and CGL2013-45410-R. The financial support for EARLINET in the ACTRIS Research Infrastructure
Project by the European Union's Horizon 2020 research and innovation programme under grant agreement no. 654169 and previously under grant agreement no. 262254 in the 7[th] Framework Programme (FP7/2007-2013) is gratefully acknowledged. This work was also supported by the University of Granada through the contract "Plan Propio. Programa 9. Convocatoria 2013" and the grant AP2009-0559. Data supporting this article is available upon request through the corresponding author.





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




**Table 1: List of functional blocks (name and Müller matrix symbol), lidar parameters (name and symbol) used for describing the lidar performance. Error source describes the parameters involved in the polarizing sensitivity.**

| Functional block | | Parameters | | Error Source |
|---|---|---|---|---|
| Name | Symbol | Name | Symbol | |
| Laser | $I_L$ | Laser intensity | $I_L$ | no |
| | | Laser depolarization parameter | $a_L$ | omitted |
| | | Misalignment angle of the polarizing plane of the laser[1] | $\alpha$ | yes |
| Emitting optics | $M_E$ | Effective diattenuation | $D_E$ | yes |
| | | Effective phase shift | $\Delta_E$ | yes |
| | | Effective misalignment angle[1] | $\beta$ | yes |
| Receiving optics | $M_o$ | Effective diattenuation | $D_o$ | yes |
| | | Effective phase shift | $\Delta_o$ | yes |
| | | Effective misalignment angle[1] | $\gamma$ | yes |
| Calibrator | $C$ | Calibration angle | $\phi$ | no |
| | | Misalignment angle[1] | $\varepsilon_r$ | yes |
| Polarizing splitter | $M_S$ | Measurement angle | $\Psi$ | no |
| | | Parallel-polarised[1] light transmittance | $D_T$ $T_p$ | yes |
| | | Perpendicular-polarised[1] light transmittance | $T_s$ | |
| | | Parallel-polarised[1] light reflectance | $D_R$ $R_p$ | yes |
| | | Perpendicular-polarised[1] light reflectance | $R_s$ | |
| | | Photomultiplier gain factor transmitted signal | $\eta_T$ | no |
| | | Photomultiplier gain factor reflected signal | $\eta_R$ | no |

[1]with respect to the PBS incident plane.





**Table 2: Values and uncertainties of the synthetic lidar parameters.**

| Parameters | | | Value | Uncertainty |
|---|---|---|---|---|
| $I_L$ | $I_0$ | | 1 | - |
| | $a_L$ | | 0 | - |
| | $\alpha$ | | 0 | $\pm 10°$ |
| $M_E$ | $D_E$ | | 0.0 | $\pm 0.2$ |
| | $\Delta_E$ | | 0° | $\pm 180°$ |
| | $\beta$ | | 0° | $\pm 10°$ |
| $M_o$ | $D_o$ | | 0.0 | $\pm 0.3$ |
| | $\Delta_o$ | | 0° | $\pm 180°$ |
| | $\gamma$ | | 0.0° | $\pm 5°$ |
| $C$ | $\phi$ | | $\pm 45°$ | - |
| | $\varepsilon_r$ | | 0° | $\pm 5°$ |
| $M_S$ | $\Psi$ | | 90° | - |
| | $D_T$ | $T_p$ | 0.95 | $\pm 0.05$ |
| | | $T_s$ | 0.01 | $\pm 0.01$ |
| | $D_R$ | $R_p$ | 0.05 | $\pm 0.05$ |
| | | $R_s$ | 0.99 | $\pm 0.01$ |
| | $\eta_R$ | | 1 | - |
| | $\eta_T$ | | 1 | - |

**Table 3: Minima and maxima of the volume linear depolarization ratio, $\delta$, set of solutions from the Monte Carlo technique applied to the synthetic lidar.**

| Lidar | $\delta = 0.004$ | | $\delta = 0.45$ | |
|---|---|---|---|---|
| | min | max | min | max |
| Synthetic | -0,01 | >1 | 0.2 | >1 |





**Table 4: EARLINET lidar systems participating in the depolarization uncertainty study.**

| Lidar name | Institution |
| --- | --- |
| LB21-IV-D200 | National Technical University of Athens, Greece |
| LB21-IV-D200 | Cyprus University of Technology, Limassol, Cyprus |
| MULHACEN | CEAMA, University of Granada, Spain |
| RALI | INOE 2000, Bucharest, Romania |
| POLLY-XT SEA | TROPOS, Leipzig, Germany |
| POLIS | LMU Munich, Germany |
| MUSA | CNR-IMAA, Potenza, Italy |
| IPRAL | IPSL/SIRTA - CNRS-Ecole Polytechnique, Palaiseau, France |





| | Property | MULHACEN 532 nm Value | Uncertainty (±) | RALI 532 nm Value | Uncertainty (±) | LB21 532 nm Value | Uncertainty (±) | IPRAL 355 nm Value | Uncertainty (±) |
|---|---|---|---|---|---|---|---|---|---|
| $I_L$ | $\alpha$ | 7.0° | 1.0° | 8° | 0.20° | 0° | 2.0° | 0° | 2.0° |
| | Included | yes | | yes | | yes | | yes | |
| $M_E$ | $D_E$ | 0.00 | 0.10 | 0 | 0.10 | 0.00 | 0.05 | 0.00 | 0.05 |
| | $\Delta_E$ | 0° | 180° | 0° | 180° | 0° | 180° | 0° | 180° |
| | $\beta$ | 0.0° | 1.0° | 0° | 1.0° | 0° | 1.0° | 0° | 1.0° |
| | $D_o$ | 0.35 | 0.04 | 0.2 | 0.10 | 0.00 | 0.05 | -0.012 | 0.012 |
| | $\Delta_o$ | 0° | 180° | 0° | 180° | 0° | 180° | 0° | - |
| | $\chi$ | 0° | 0.5° | 0° | 0.5° | 0° | 0.5° | 0° | 0.5° |
| $M_o$ | Location (in front of) | $\mathbf{M_o}$ | | $\mathbf{M_o}$ | | $\mathbf{M_s}$ | | $\mathbf{M_s}$ | |
| | Element | Polarizer | | Polarizer | | Rotator | | 0-order waveplate | |
| | $\varepsilon$ | 0.00° | 0.10° | 0° | 0.1° | 0° | 0.1° | 0° | 0.1° |
| $C$ | Type | PBS | | PBS | | PBS | | PBS + polarizers | |
| $M_S$ | Polarizing components | Parallel + perpendicular | | Parallel + perpendicular | | Parallel + perpendicular | | Parallel + perpendicular | |
| | $T_p$ | 0.95 | 0.01 | 0.99 | 0.01 | 0.95 | 0.01 | 1 | - |
| | $T_s$ | 0.005 | 0.001 | 0.001 | 0.001 | 0.01 | 0.01 | 0 | - |
| | $R_p$ | $1-T_p$ | $\Delta T_p$ | $1-T_p$ | $\Delta T_p$ | $1-T_p$ | $\Delta T_p$ | 0 | - |
| | $R_s$ | $1-T_s$ | $\Delta T_s$ | $1-T_s$ | $\Delta T_s$ | $1-T_s$ | $\Delta T_s$ | 1 | - |

**Table 5:** Values and uncertainties of each lidar parameter for the set of simulated EARLINET lidars. $\Delta T_p$ and $\Delta T_s$ are the $T_p$ and $T_s$ uncertainties.





| Property | | MUSA 532 nm Value | MUSA 532 nm Uncertainty (±) | POLLY-XT SEA 532 nm Value | POLLY-XT SEA 532 nm Uncertainty (±) | POLIS 355nm/532nm 355 nm Value | POLIS 355nm/532nm 355 nm Uncertainty (±) | POLIS 355nm/532nm 532 nm Value | POLIS 355nm/532nm 532 nm Uncertainty (±) |
|---|---|---|---|---|---|---|---|---|---|
| $I_L$ | $\alpha$ | 3.0° | 0.6° | 0.0° | 1.0° | 0.0° | 0.5° | 0.0° | 0.5° |
| | Included | no | | yes | | no | | | |
| $M_E$ | $D_E$ | - | - | 0.00 | 0.10 | - | - | - | - |
| | $\Delta_E$ | - | - | 0 | 180° | - | - | - | - |
| | $\beta$ | - | - | 0.0° | 1.0° | - | - | - | - |
| $M_O$ | $D_O$ | -0.055 | 0.003 | 0.011 | 0.022 | -0.001 | 0.001 | 0.011 | 0.011 |
| | $\Delta_O$ | 0° | 180° | 0° | 180° | 0° | 180° | 0° | 180° |
| | $\chi$ | 0° | 0.1° | 0.0° | 0.5° | 0.00° | 0.10° | 0.00° | 0.10° |
| C | Location (in front of) | $M_S$ | | $M_O$ | | Rotator | | $M_o$ | |
| | Type | 0-order waveplate | | Polarizer | | | | | |
| | $\varepsilon$ | 1.15° | 0.3° | 3" | 3" | 0.0° | 1.0° | 0.0° | 1.0° |
| | Type | PBS + polarizers | | Polarizer | | PBS + polarizers | | | |
| | Polarizing components | Parallel + perpendicular | | Total + perpendicular | | Parallel + perpendicular | | | |
| $M_S$ | $T_p$ | 1 | - | 0.532 | 0.017 | 0.21 | - | 0.79 | - |
| | $T_s$ | 0 | - | 0.500 | 0.015 | 0 | - | 0 | - |
| | $R_p$ | 0 | - | $ER(1-T_p)$ | - | 0 | - | 0 | - |
| | $R_s$ | 1 | - | 0.5 | 0.015 | 0.225 | - | 0.8 | - |

**Table 5 (continuation):** Values and uncertainties of each lidar parameter for the set of simulated *EARLINET* lidars. *ER* is the linear-polarizer extinction ratio.





1 **Table 6: Minima and maxima of the simulated $\delta$ for the EARLINET lidar systems at $\delta = 0.004$ and $\delta = 0.45$.**
2 **$\Delta\delta$ is the min and max range. Decimal round performed according to the first non-zero standard-deviation digit**
3 **(not shown).**

| Lidar | | $\delta = 0.004$ | | $\delta = 0.45$ | |
|---|---|---|---|---|---|
| | | min | max | min | max |
| MULHACEN | | -0,012 | 0,039 | 0,437 | 0,477 |
| RALI | | -0,012 | 0,034 | 0,436 | 0,474 |
| LB21 | | -0,006 | 0,024 | 0,399 | 0,512 |
| IPRAL | | 0,0039 | 0,0098 | 0,4393 | 0,4654 |
| MUSA | | -0,0011 | 0,0066 | 0,4431 | 0,4548 |
| POLLY-XT SEA | | 0,0039 | 0,0096 | 0,4446 | 0,4602 |
| POLIS | 355nm | 0,004 | 0,0049 | 0,45 | 0,4507 |
| | 532nm | 0,004 | 0,0049 | 0,45 | 0,4507 |



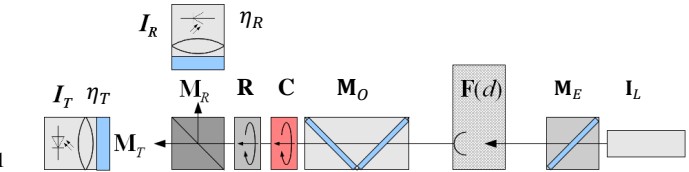

2    **Figure 1: Lidar scheme based on functional blocks (adapted from Freudenthaler (2016a)).**





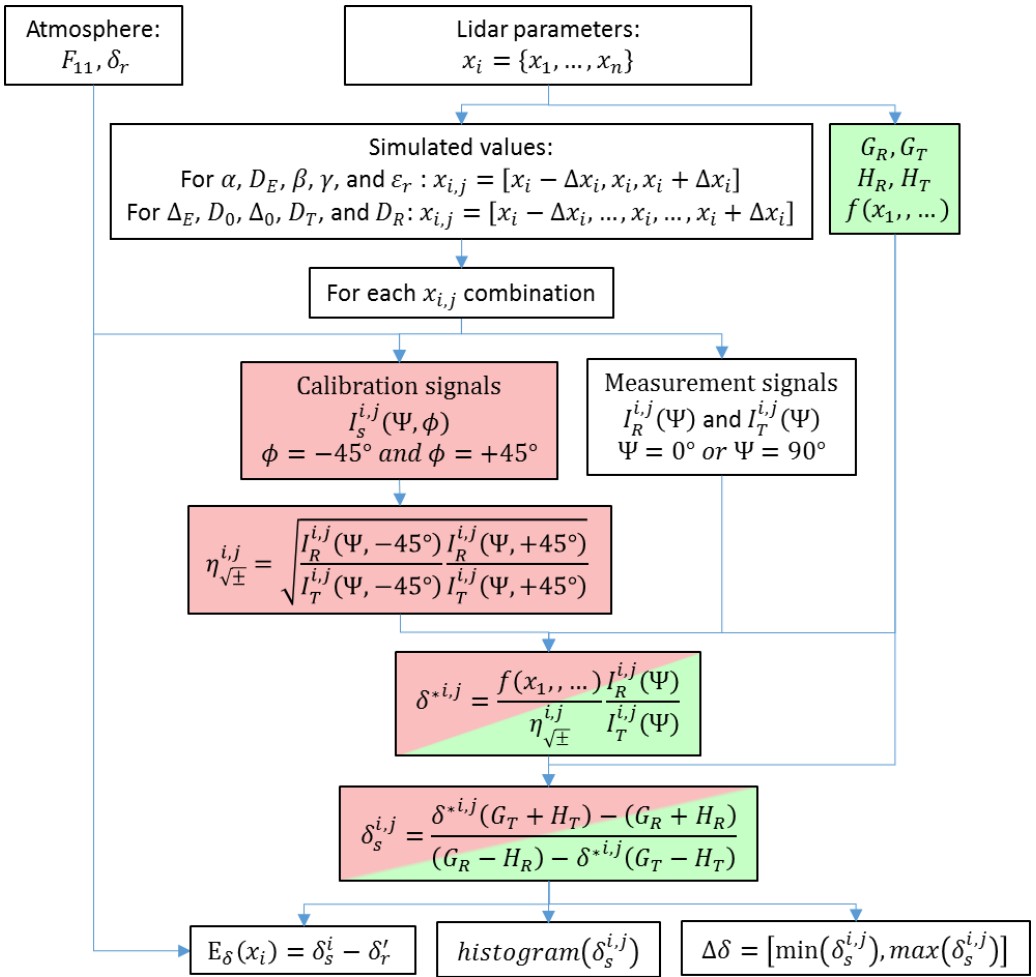

2      **Figure 2: PLS diagram flux. Depolarization calibration steps are marked in red whereas the correction ones**
3      **applied thanks to the known lidar parameters (lidar polarizing sensitivity characterization) are marked in**
4      **green. $x_1, \ldots, x_n$ are the lidar parameters from Table 1.**





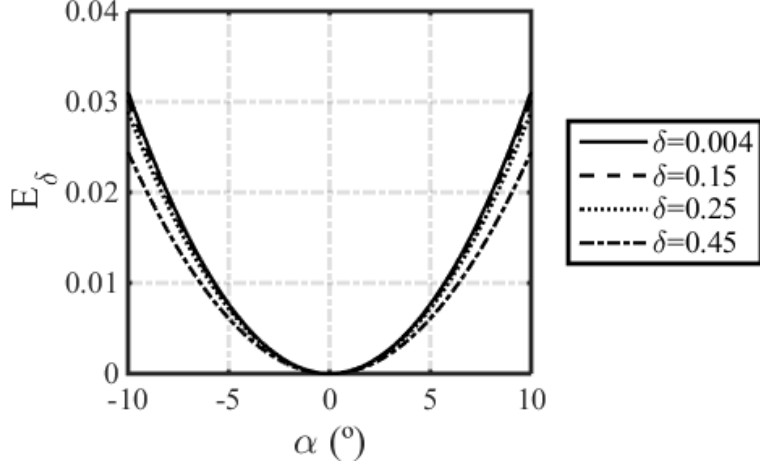

2    **Figure 3:** $E_\delta$ **depending on** $\alpha$ **for different** $\delta$ **values.**





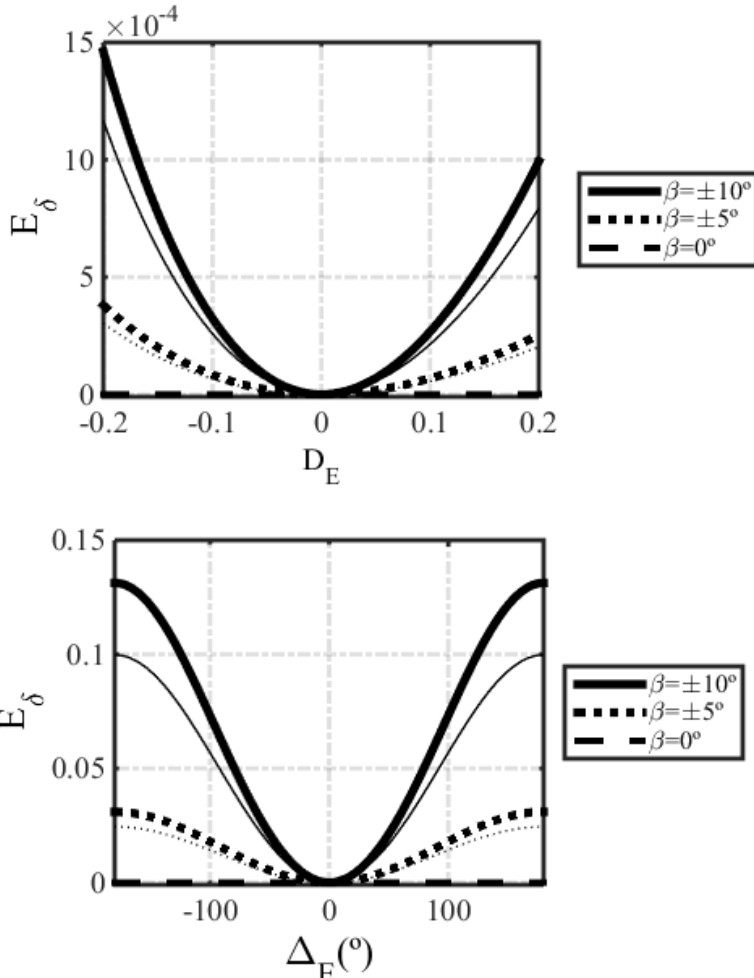

**Figure 4:** $E_\delta(D_E, \beta)$ dependence on $D_E$ (top) parameterized by $\beta$ and $E_\delta(\Delta_E, \beta)$ dependence on $\Delta_E$ (bottom). Thick
and thin lines correspond to $\delta$ values of 0.004 and 0.45, respectively.





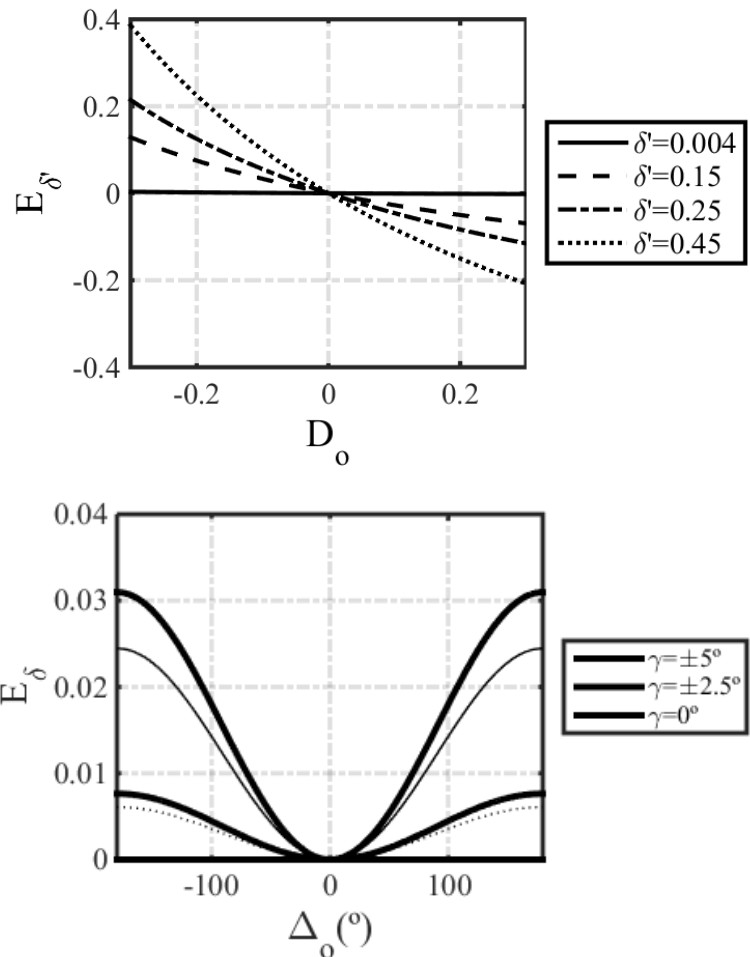

2 **Figure 5:** $E_\delta(D_o)$ **parametrizing** $\delta$**(top).** $E_\delta(\Delta_o, \gamma)$ **parameterizing** $\gamma$ **(bottom). Thick and thin lines correspond to**
3 $\delta$ **values of 0.004 and 0.45, respectively.**




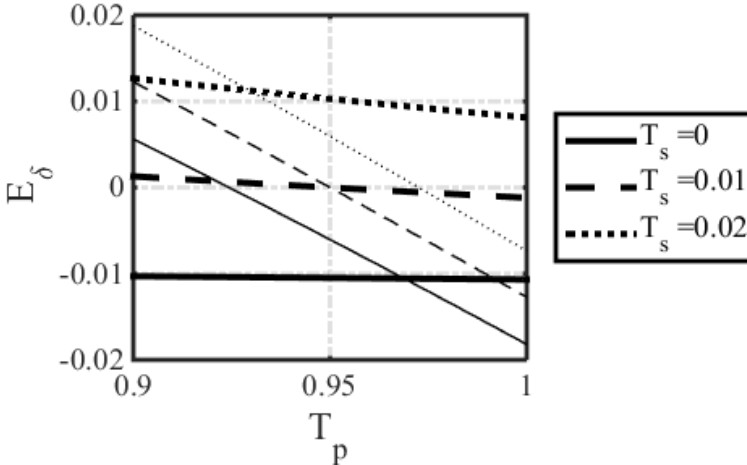

2 **Figure 6:** $E_\delta$ **depending on** $T_p$ **parameterizing** $T_s$**. Thick and thin lines correspond to** $\delta$ **values of 0.004 and 0.45,**

3 **respectively.**

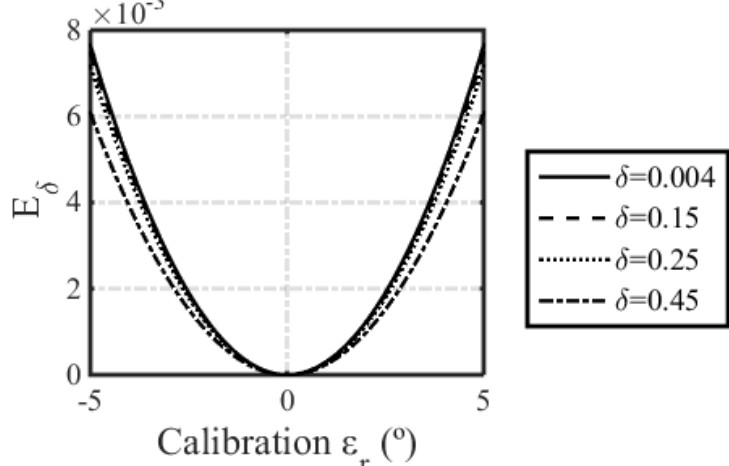

6 **Figure 7:** $E_\delta$ **depending on** $\varepsilon_r$ **parameterizing** $\delta$ **according to the label.**



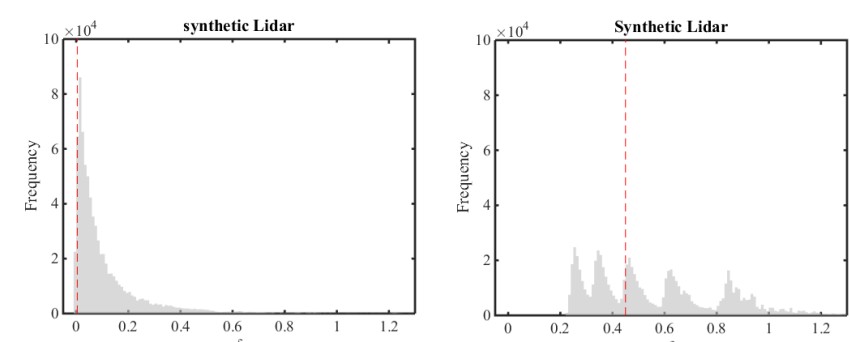

2    **Figure 8: $\delta$ histogram for the synthetic lidar. Dased line represent the real $\delta$, $\delta_r$, at** 0.004 **(left) and** 0.45 **(right).**





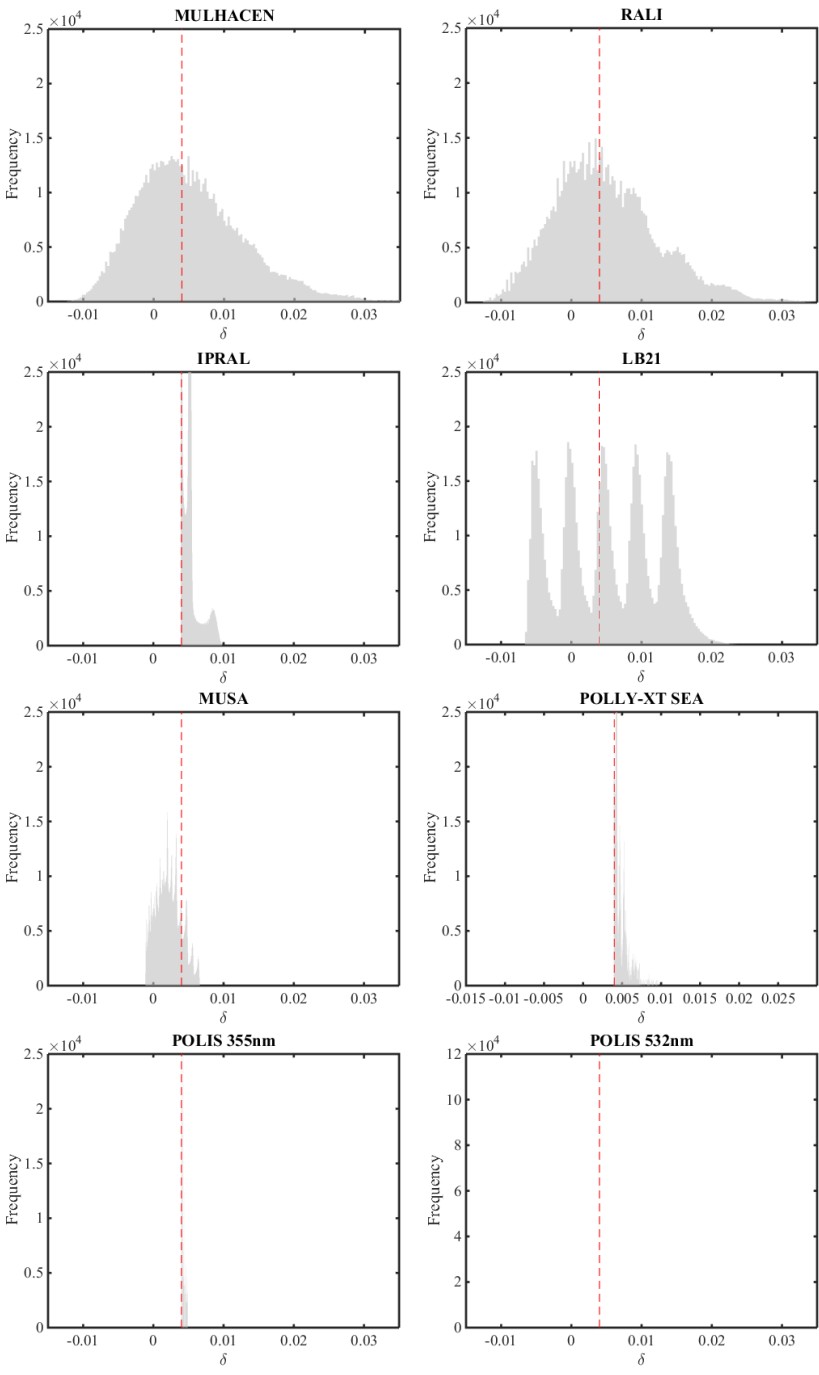

2      **Figure 9: $\delta$ histogram for several EARLINET lidars (Table 4). Dashed lines represent the real $\delta$, $\delta_r$, at 0.004.**





1    **Figure 10: δ histogram for several EARLINET lidars (Table 4). Dashed lines represent the real δ, $δ_r$, at** 0.45.





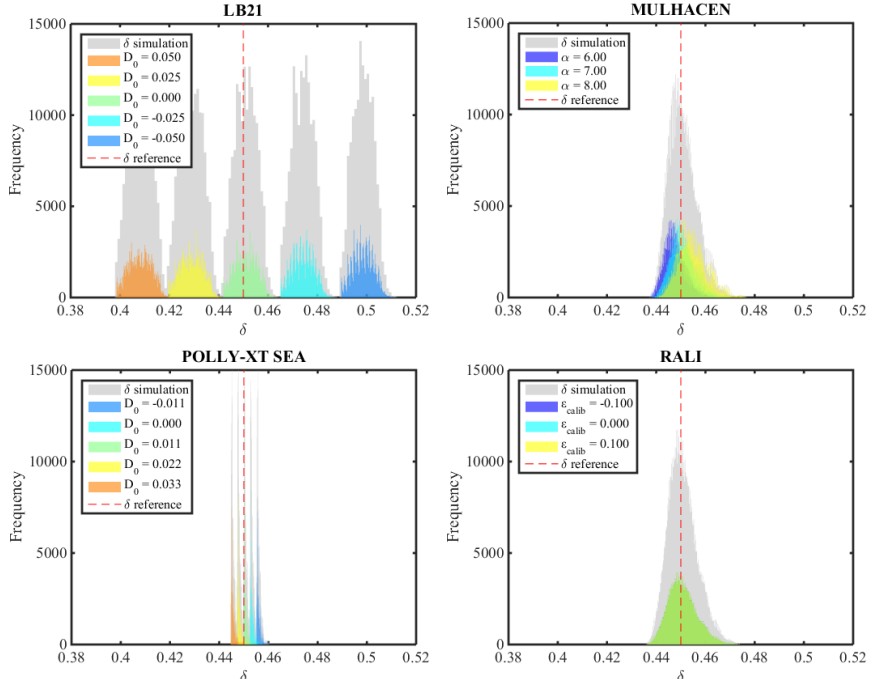

1  **Figure 11:** $\delta$ **sub-histograms for** $\alpha$, $D_o$ **and** $D_T$ **values, according to the labels, for the simulation of LB21,**
2  **MULHACEN, POLLY-XT SEA and RALI. Dashed lines represent the reference** $\delta$, $\delta_r$, **at** 0.45.

