# Peer review of "Assessment of lidar depolarization uncertainty by means of a polarimetric lidar simulator"

_Atmospheric Measurement Techniques, 2015_

## Referee Comment (RC1) · F. Cairo (Referee) · 18 Mar 2016

In the elastic lidar technique, the assessment of the degree of depolarization induced by particle backscattering of polarized light, is a key factor to discriminate between different particle shapes. Unfortunately, its evaluation may suffer from inaccuracy, strongly depending on the lidar system setup. Thus, this interesting study on the sensitivity and accuracy of lidar depolarization measurements is contributing to define common procedures of lidar calibration, and surely deserve publication on the journal. Its aim is a quantification of the volume linear depolarization ratio (a common parameter measured by lidars) uncertainty due to systematic errors. It also presents a software tool (Polarimetric Lidar Simulator (PLS) applied to synthetic as well as real li-

dar systems to quantify the uncertainties and inaccuracies induced by the various lidar subsystems. The main outcome of the paper is the identification of which instrumental parameters need more accurate characterization and it then helps giving guidance in the development of new lidar systems with better performance in depolarization measurements, and in the standardization of their products. I recommend its publication. However, there are a some minor issues that I would like the author to look at.

In the following, (page, lines).

(3,13) To my knowledge, laser commonly used in lidar practice are often guaranteed with linear polarization not better than 100:1, so, as reported, sometimes a polarizing cube is used to further purify the laser light polarization before the transmission into the atmosphere. However, this does not prevent problems arising from possible misalignments between the laser polarizing plane and the polarizing splitter incident plane. Maybe it is worthwhile noting that the two effects are inherently different, and can in principle be corrected differently, in one case by further filtering the laser light to remove the unpolarized residuals, in the other, by a proper alignment of the two (polarization and incident) planes. However, the authors' formalism is correct, and general.

(7,5-6) This claim should be substantiated or referenced.

(7,18-20) What follows is the crucial point of my review. I think that the sentence reported in the text understates what, to my opinion, is one cause of concern about the accuracy of all the absolute calibration techniques of the lidar signal which have been proposed so far. The reference to Bravo-Aranda et al., refers to the stability of photomultiplier gains over long times, but I think it is not sufficient to guarantee that. I hope that, if I behave myself, after my departure I will find myself in a place where the sensors' responses to signals are linear along their entire dynamic range. Unfortunately, it is well known that photomultipliers are far from heaven, both when used in photoncounting mode, or in current mode. The lidar return may be a more or less significant part of the total signal detected by the photomultiplier, depending on the altitude where

it originates and, more significantly, on the sky background that can vary over several orders of magnitude. This means that the whole lidar returns are located on different portions of the photomultiplier response curve, in dependence of the sky background. These different portions can be locally linear, or quasi-linear, but may not share the same linearity. In other words, depending on the sky background, the photoimpulse height spectrum of a photomultiplier can change substantially, thus affecting both photoncounting and current mode of detection. This effect may be dramatic or negligible, depending on the photomultiplier type, polarization, single realization of the device, and so on. Of course, if the effect is there, it has an impact on the absolute calibration, that became dependent on the sky background conditions. I am not aware of any study that focused on the dependency of the absolute calibration on sky background conditions, but in my experience as a researcher, I saw changes of volume depolarization values by few percents, simply induced by the sun rising or setting, so I am quite sure this effect can be present, even if it can be reduced or suppressed by an accurate choice of the photomultiplier type, polarization and amplification circuitry, lidar spectral bandwith and so on. I am not saying that this effect is spoiling the results of this study, or the whole absolute calibration procedures. What I am saying is that the assumption of a constant photomultiplier gain is quite a severe one, and should be acknowledged as that.

(9,5) The relationship between depolarization ratio and particle "asphericity" (whatever that means) is not so straightforward. Even under the simplified assumption of particles as oblate or prolate spheroids - unrealistic, but widely used because it allows analytical expression for the scattering equations solution - one could find greater depolarization ratio for aspect ratioes close to unity. The authors might drop that sentence, or quote some reference to T-matrix computations, as instance.

(12,6-7) "do not use laser emitting optics if possible" please rephrase, as in this form, it is not clear what it is meant, at first sight.

(16,3) It may be worthwhile to note here that the opposite result applies when a relative calibration approach is pursued, i.e. when the theoretical value of the molecular depolarization is imposed in a region of the lidar profile which is free of aerosol. In that case, the instrumental effects here discussed lead to an underestimation of the aerosol depolarization.

———————————————

---

## Short Comment (SC1) · 6 Apr 2016

Let me please seize the opportunity to participate to the open AMT-discussion on the interesting manuscript proposed by Bravo-Aranda et al.. This manuscript is interesting as it deals with the important concern of performing sensitive and accurate depolarization lidar measurements and proposes to use a simulator to address these concerns. I have several questions on this manuscript, which I hope may benefit to the authors and to potential readers.

1. In their manuscript, Bravo-Aranda et al. indicate that they propose a new tool to assess the lidar polarizing sensitivity and to quantify the error on the depolarization due to the unknown systematic errors. To my knowledge, such a discussion on the

sensitivity and accuracy of polarization lidar measurements was proposed in (G. David et al., Appl. Phys B, 108, 197-216, 2012) through laboratory and field experiments, by quantitatively discussing all possible systematic errors, including the following ones that are here analyzed: a small-unpolarized component in the emitter laser beam, an imperfect separation of polarization components, a misalignment between the transmitter and receiver polarization axes. Could the authors situate their work in regards to other published works and identify the novelty of their work ? I feel quite surprised that this paper is not quoted.

2. As a reader, it seems to me that the term of "polarizing sensitivity ", as proposed all along the manuscript, needs some clarification. Could the authors add precision ?

3. The Stokes-Mueller matrix formalism is here used by introducing the Mueller matrix of the emitter optics and of the receiver optics with reference to Freudenthaler (2016a) all along the manuscript, a reference that is not yet published and hence not yet reviewed. For the sake of clarity, I propose to the authors to use reference to the publication by G. David et al. (Opt. Exp., 21, 16, 18624-18639, 2013) where with my co-authors, we stated the Stokes-Mueller matrix formalism for a pulsed laser source in the lidar backscattering geometry by introducing the same emitting and receiver optics Mueller matrices and the corresponding formalism. Could the authors situate their work in regards to this published work? I feel quite surprised that this paper be not quoted.

4. When reading the manuscript, the reader gets the impression that the approach that is here proposed (the polarization lidar simulator) is the only possible one to address the sensitivity and the accuracy of lidar depolarization measurements. In David et al. (APB, 2012), we used a somewhat different approach by addressing the detector transfer matrix, which allows, as being diagonal, to perform a robust calibration, after minimizing all possible effects on the emitter optics. What additional information(s) do the authors get by introducing the Stokes-Mueller matrix formalism compared to our contribution? Whether this formalism is necessary or not should be clearly stated for

potential readers.

5. The manuscript only deals with $1\lambda$-polarization lidars while many papers have been published with $2\lambda$-polarization lidars such as Shimizu et al., 2004 or Tesche et al., 2009, as well-known. Such polarization lidar stations rely on wavelength-dependent components, that may attenuate the backscattering intensity and also modify the polarization state of the backscattered radiation in agreement with Fresnel's formulas. How do the authors account for this important contribution ? Is their conclusion similar to that published in David et al. (APB, 2012) ? Different ?

6. Using a polarization simulator may be interesting but it is not to my knowledge the only possible approach. To address accurate lidar depolarization measurements or to calibrate the lidar depolarization, laboratory experiments at exact $180°$ are nowadays available and can be used to quantitatively address this quantity, even for non-spherical particles such as mineral dust particles, and at two-wavelengths, as we recently published (Miffre et al., JQSRT 2016). I think adding such information may be useful for potential readers of your manuscript, as it complements your approach. Indeed, a robust calibration has to rely on accurate laboratory measurements. That's why, I am proposing to add this reference that complements your work.
* * *

---

## Referee Comment (RC2) · Anonymous Referee #1 · 25 Apr 2016

Theoretical assessment of the lidar polarizing sensitivity

Bravo-Aranda &al.

A good paper, each step of the theoretical modeling is reasonably well explained except the definition of the Stokes vector used and consequently the corresponding Muller matrices.

Following Del Guasta et al. [1]; most of polarization lidars use linearly polarized light and a linear depolarization ratio defined as

$$\delta_{lin} = \frac{I_\perp}{I_{//}} = \frac{S_0 - S_1}{S_0 + S_1} \quad . \qquad (1)$$

To define circular depolarization, we have to take into account the fact that light with a left circular polarization incident on a spherical particle is changed to right circular polarization when it is backscattered. In comparison, for an incident linearly polarized light, the polarization state is preserved although the electric field sign is changed. To maintain a definition consistency between the linear and circular depolarization ratios, the depolarization ratios must be a measure of the backscattered signal polarization departure. So, the circular depolarization in conformity with [2] by using

$$\delta_{cir} = \frac{I_{C\perp}}{I_{C//}} = \frac{S_0 + S_3}{S_0 - S_3} \quad . \qquad (2)$$

However, as stated in Eq. 2.23, it is the depolarization parameter that is truly a measure of the depolarisation caused by the aerosol independently of the polarization state (linear or circular) of the lidar. I understand depolarisation ratio measurement inherited from strong a legacy; at the minimum a reference to Gimmestad [3] (see at the bottom a list of pertinent references) and a short paragraph explaining how to transform depolarisation ratio to depolarisation parameter is required. Your definition of the Muller matrix for randomly oriented is inconsistent with the one use the references above:

$$M_{atm} = p(180^o) \begin{pmatrix} 1 & 0 & 0 & 0 \\ 0 & 1-d & 0 & 0 \\ 0 & 0 & d-1 & 0 \\ 0 & 0 & 0 & 2d-1 \end{pmatrix}$$

Unless 'a' the polarisation parameter is define as '1-d', 'd' being the depolarisation parameter. It needs to be clarified.

The work and references are highly EARLINET center:

On the effect of mirror on depolarization measurement reference should be made to Bissonnette [4].

On measurements techniques that cancel out most of system depolarisation artifacts, reference to Cao [5] should be of interest.

In the following some specific recommendations;

The figure caption for figure 1 is anemic; identify each component; if I understood correctly, R should be identify as a lambda/2 waveplate in the text and in the figure caption.

In eq 2.1, why 2 alpha instead of alpha.

For Eq. 2.2 a reference is required.

In Eq. 2.13, why 2 beta instead of beta

For Eq 2.23, it should be specify it is the scattering matrix for randomly oriented particle; reference to Michenko [2], Gimmestad [3] and Roy [6] should be made;

Eq. 2.27, how Gs and Hs are obtained?

The authors should know that the average reader what to have a good idea of the meaning of a graphic by simply reading the caption. So for figure 3, 4, 5, 6, 7 and 8 spell out clearly all the meaning of variables. It is important.

Recommended references

[1]. M. Del Guasta. et al., "Use of polarimetric lidar for the study of oriented ice plates in clouds," Appl. Opt. **45**, pp. 4878-4887 (2006).

[2]. M.I. Mishchenko, J.W. Hovenier, "Depolarization of light backscattered by randomly oriented nonspherical particles," Optics Letters 20, pp.1356-1358 (1995).

[3]. G. G. Gimmestad, "Reexamination of depolarization in lidar measurements," *Appl. Opt.* 47, 3795-3802 (2008).

[4]. L.R. Bissonnette, G. Roy and F. Fabry, "Range-height scans of lidar depolarization for characterizing the phase of clouds and precipitation," J. Atmos. and Oceanic Tech., 18, 1429-1446 ( 2001).

[5]. X. Cao, G. Roy and R. Bernier, "Lidar polarization discrimination of bioaerosols," *Opt. Eng.* *49(11) 116201-1-12* (2010).

[6]. G. Roy , X. Cao, and R. Bernier, "On linear and circular depolarization lidar signatures in remote sensing of bioaerosol – experimental validation of the Mueller matrix for randomly oriented particles," *Opt. Eng. xx(11) 116201-1-12* (2011).

---

## Author Comment (AC1) · 7 Jul 2016

Reply to Referee 1

The authors would like to thank Dr. Cairo for their thoughtful and helpful comments and suggestions. His review has made a significant contribution to the improvement of the paper. The line numbering in the reviewers' comments refers to the manuscript published in AMTD whereas the line numbering in the responses refers to the new version of the manuscript.

**Comment**: (3,13) To my knowledge, laser commonly used in lidar practice are often guaranteed with linear polarization not better than 100:1, so, as reported, sometimes a polarizing cube is used to further purify the laser light polarization before the transmission into the atmosphere. However, this does not prevent problems arising from possible misalignments between the laser polarizing plane and the polarizing splitter incident plane. Maybe it is worthwhile noting that the two effects are inherently different, and can in principle be corrected differently, in one case by further filtering the laser light to remove the unpolarised residuals, in the other, by a proper alignment of the two (polarization and incident) planes. However, the authors' formalism is correct, and general.

**Answer:** We agree that the problem of the "polarization purity" of the laser should be discussed in more detail, Furthermore, we assume that the main contributor to a possible elliptical polarization of the emitted laser beam is the emitter optics and neglect the contribution of the laser itself. We changed the paragraph (page 3, lines 8-17) as follows:

The specified "polarization purity" of lasers commonly used in lidars is on the order of 100:1, if it is specified at all. Already the terminology indicates that such specifications are rather vague. Actually, laser manufacturers do not measure the state of polarization of the laser beams, and seem to give values which are under all circumstances on the safe side. Theoretically, nonlinear crystals as second and third harmonic generators should provide very clean linear polarization, just depending on the quality and accuracy of alignment of the crystals. Due to lack of detailed information we neglect this errors source in this work. However, in order to remove this uncertainty, in some lidar systems high quality polarizing beam splitters are used to improve the degree of linear polarization of the emitted laser beams. In both cases, the plane of polarization of the laser beam can be rotated by angle α with respect to the incident plane of the polarizing beam splitter in the receiver optics, which results in the Stokes vector $I_L$ of the emitted laser beam:

**Comment**: (7,5-6) This claim should be substantiated or referenced.

**Answer:** The explanation in the original manuscript was based on the wrong parameter. It is not the combined retardation of the cleaned PBS that is the main error source, because it is assumed that the extinction ratio of the cleaning polarizing sheet filters is sufficiently small so that the cross-talk can really be neglected. However, in case the polarizing sheet filters are not aligned well with the PBS, their effect is reduced. We rephrased the paragraph (page 7, lines 10-12) accordingly:

We assume that the extinction ratio of the cleaning polarizing sheet filters is sufficiently small and that they can be oriented with an accuracy much better than ±5° with respect to the PBS, and that therefore the resulting error of $D_S^{\#}$ can be neglected (See Appendix).

and added and appendix (page 20) which quantifies the reduction and the consequential required alignment accuracy.

From the Müller matrix of two rotated diattenuators (see Freudenthaler, 2016; supplement Eqs. S.10.10.1)

$$\frac{\langle M_A(\phi) M_S|}{T_A T_S} = \langle 1 + c_{2\phi} D_A D_S \quad D_S + c_{2\phi} D_A \quad s_{2\phi} D_A Z_S c_S \quad s_{2\phi} D_A Z_S s_S|$$

we get the p- and s-polarised transmissions $T^p$ and $T^s$

$$\frac{T_{AS}^p}{T_A T_S} = \left(1 + c_{2\phi} D_A D_S\right) + \left(D_S + c_{2\phi} D_A\right) = \left(1 + D_S\right)\left(1 + c_{2\phi} D_A\right)$$

$$\frac{T_{AS}^s}{T_A T_S} = \left(1 + c_{2\phi} D_A D_S\right) - \left(D_S + c_{2\phi} D_A\right) = \left(1 - D_S\right)\left(1 - c_{2\phi} D_A\right)$$

and the extinction ratio $\rho_{AT}$ for the transmitted path

$$\rho_{AT} = \frac{T_{AT}^s}{T_{AT}^p} = \frac{\left(1 - D_T\right)\left(1 - c_{2\phi} D_A\right)}{\left(1 + D_T\right)\left(1 + c_{2\phi} D_A\right)} = \rho_T \frac{1 - c_{2\phi} D_A}{1 + c_{2\phi} D_A} = \rho_T \frac{\left(T_A^p + T_A^s\right) - c_{2\phi}\left(T_A^p - T_A^s\right)}{\left(T_A^p + T_A^s\right) + c_{2\phi}\left(T_A^p - T_A^s\right)}$$

$$= \rho_T \frac{\left(1 + \rho_A\right) - c_{2\phi}\left(1 - \rho_A\right)}{\left(1 + \rho_A\right) + c_{2\phi}\left(1 - \rho_A\right)} = \rho_T \frac{1 - c_{2\phi} + \rho_A\left(1 + c_{2\phi}\right)}{1 + c_{2\phi} + \rho_A\left(1 - c_{2\phi}\right)} = \rho_T \frac{tan^2\phi + \rho_A}{1 + \rho_A tan^2\phi}$$

$$\approx \rho_T \rho_A \left(1 + \frac{tan^2\phi}{\rho_A}\right)$$

Please note that p- and s-polarisations are with respect to the incidence plane of the polarising beam-splitter cube MS and that the polarising sheet filter is rotated by an angle of 90° in the reflected path, with $\cos(2 \cdot (90° + \phi)) = \cos(2\phi)$. The extinction ratio $\rho_{AR}$ for the reflected path can be derived in the same way

$$\rho_{AR} = \frac{T_{AR}^s}{T_{AR}^p} = \rho_R \frac{tan^2\phi + \rho_A}{1 + \rho_A tan^2\phi} \approx \rho_R \rho_A \left(1 + \frac{tan^2\phi}{\rho_A}\right)$$

For a typical polarising beamsplitter cube with $T_T^p = 0.95$, $T_R^s = 0.99$ and $T_T^s = 1 - T_R^s$, $T_R^p = 1 - T_T^p$, the extinction ratio in the transmitted path is $\rho_T = T_T^s/T_T^p = 0.0105$ and in the reflected path $\rho_R = T_R^s/T_R^p = 0.0505$. Using additional cleaning polarising sheet filters with $\rho_A = 0.01$, the combined extinction ratios $\rho_{AS} = \rho_A \cdot \rho_S$ are improved by a factor of 100. A misalignment of the polarising sheet filter by an angle $\phi$ with $tan^2\phi = \rho_A$, for example, decreases the improvement by a factor of two, which is in this case about $\phi = 5.7°$ (for $\rho_A = 0.01$).

**Comment**: (7,18-20) What follows is the crucial point of my review. I think that the sentence reported in the text understates what, to my opinion, is one cause of concern about the accuracy of all the absolute calibration techniques of the lidar signal which have been proposed so far. The reference to Bravo-Aranda et al., refers to the stability of photomultiplier gains over long times, but I think it is not sufficient to guarantee that. I hope that, if I behave myself, after my departure I will find myself in a place where the sensors' responses to signals are linear along their entire dynamic range. Unfortunately, it is well known that photomultipliers are far from heaven, both when used in photoncounting mode, or in current mode. The lidar return may be a more or less significant part of the total signal detected by the photomultiplier, depending on the altitude where it originates and, more significantly, on the sky background that can vary over several orders of magnitude. This means that the whole lidar returns are located on different portions of the photomultiplier response curve, in dependence of the sky background. These different portions can be locally linear, or quasi-linear, but may not share the same linearity. In other words, depending on the sky background, the photo impulse height spectrum of a photomultiplier can change substantially, thus affecting both photoncounting and current mode of detection. This effect may be dramatic or negligible, depending on the photomultiplier type, polarization, single realization of the device, and so on. Of course, if the effect is there, it has an impact on the absolute calibration, that became dependent on the sky background conditions. I am not aware of any study that focused on the dependency of the absolute calibration on sky background conditions, but in my experience as a researcher, I saw changes of volume depolarization values by few percents, simply induced by the sun rising or setting, so I am quite sure this effect can be present, even if it can be reduced or suppressed by an accurate choice of the photomultiplier type, polarization and amplification circuitry, lidar spectral bandwidth and so on. I am not saying that this effect is spoiling the results of this study, or the whole absolute calibration procedures. What I am saying is that the assumption of a constant photomultiplier gain is quite a severe one, and should be acknowledged as that.

**Answer**: we completely agree with the referee about the importance of the linearity of the photomultiplier response and the possible effects on the signals and the calibration. We also conclude with the need for investigating each lidar setup to exclude, or minimize and characterize such non-linearities.

Because our manuscript deals with the polarization dependent errors caused by the lidar emitter and receiver optics, it is not the place to additionally deal with electronic errors. We replace the sentence accordingly (page 7 lines 20-26):

The reflected and transmitted signals are detected by the photomultipliers which perform the light to electrical signal conversion. They affect the depolarization measurements as, in general, different photomultipliers have different gains. Regarding the Stokes-Müller formalism, we define the opto-electronic gains $\eta_R$ and $\eta_T$ for the photomultiplier gains of the transmitted and reflected signals including all optical attenuation of the lidar system in the transmitted and reflected path that is independent of polarization. We set them equal to 1 since we only investigate the polarization dependent errors of the lidar optics.

**Comment**: (9,5) The relationship between depolarization ratio and particle "asphericity" (whatever that means) is not so straightforward. Even under the simplified assumption of particles as oblate or prolate spheroids - unrealistic, but widely used because it allows analytical expression for the scattering equations solution - one could find greater depolarization ratio for aspect ratioes close to unity. The authors might drop that sentence, or quote some reference to T-matrix computations, as instance.

**Answer**: the sentence has been removed.

**Comment**: (12,6-7) "do not use laser emitting optics if possible" please rephrase, as in this form, it is not clear what it is meant, at first sight.

**Answer**: the phrase has been rewritten (page 12, line 14).

It is recommended to emit the laser beam directly to the atmosphere to avoid this error source

**Comment**: (16,3) It may be worthwhile to note here that the opposite result applies when a relative calibration approach is pursued, i.e. when the theoretical value of the molecular depolarization is imposed in a region of the lidar profile which is free of aerosol. In that case, the instrumental effects here discussed lead to an underestimation of the aerosol depolarization.

**Answer**: We don't want to consider a relative calibration at all in this manuscript, because the involved uncertainty is difficult, if not impossible, to estimate accurately enough and in general unacceptably high. Furthermore, it can be seen from Figs. 9 and 10 that in some cases the distributions are shifted to lower values. Therefore we think that the limited number of lidar set-ups considered in this manuscript does not allow to state a general overestimation and removed the corresponding sentences.

---

## Author Comment (AC2) · 7 Jul 2016

Reply to Anonymous Referee #1

The authors would like to thank the referee for his thoughtful and helpful comments and suggestions. His review has made a significant contribution to the improvement of the paper. Comments of the referee are blue and answers are violet. Text in green square are included in the reviewed manuscript. The line numbering in the reviewers' comments refers to the manuscript published in AMTD whereas the line numbering in the responses refers to the new version of the manuscript.

**Comment:** […] However, as stated in Eq. 2.23, it is the depolarization parameter that is truly a measure of the depolarisation caused by the aerosol independently of the polarization state (linear or circular) of the lidar. I understand depolarisation ratio measurement inherited from strong a legacy; at the minimum a reference to Gimmestad [3] (see at the bottom a list of pertinent references) and a short paragraph explaining how to transform depolarisation ratio to depolarisation parameter is required. Your definition of the Muller matrix for randomly oriented is inconsistent with the one use the references above:

$$M_{atm} = p(180°)\begin{pmatrix} 1 & 0 & 0 & 0 \\ 0 & 1-d & 0 & 0 \\ 0 & 0 & d-1 & 0 \\ 0 & 0 & 0 & (2d-1) \end{pmatrix}$$

Unless 'a' the polarisation parameter is define as '1-d', 'd' being the depolarisation parameter. It needs to be clarified.

**Answer:** We agree that the use of the polarization parameter a instead of the de-polarization parameter d as in Gimmestadt, 2008, can cause confusion. Therefore, we add the following sentence (page 8-9 lines 33-2):

Please note that instead of the polarization parameter *a* different but equivalent expressions are used in other publications as described in more detail in Freudenthaler, 2016a. Probably most known is the de-polarization parameter *d = 1- a* used in Gimmestadt, 2008.

**Comment**: The work and references are highly EARLINET centre: On the effect of mirror on depolarization measurement reference should be made to Bissonnette [4]. On measurements techniques that cancel out most of system depolarisation artefacts, reference to Cao [5] should be of interest.

**Answer**: Bissonnette et al, 2001, use two scanning mirrors in the emitter optics and show that they indeed have an important influence on the measured depolarization ratio. We thank the reviewer for the hint to this publication and add the following sentence (page 5 lines 22-23):

The possible effect of 45°-tilted scanning mirrors on depolarization measurements was highlighted by Bissonnette et al., 2001.

Cao et al., 2010, employ a rotating half-wave plate and a quarter-wave plate in the emission optics to subsequently change the state of polarization of the emitted laser beam. They show that when measuring the same atmospheric volume with a horizontal and a vertical polarized laser beam, the two measurements together can compensate some errors. This setup is not very common, and it can introduce additional errors (e.g. temporal atmospheric changes) in the determination of the depolarization ratio. No systematic error assessment is presented there. In this manuscript we cannot include all the possibilities to measure the linear depolarization ratio and all the possible errors, but focus on the error calculation for the most commonly used types of lidars. To our opinion, the lidars used in EARLINET and those which are investigated in this manuscript are typical for the majority of lidars used in other networks. However, Cao et al., 2010, is referenced in the compagnion manuscript (Freudenthaler, 2016a),

which describes the theoretical basis of this manuscript.

Furthermore, the uncertainties in the compensation of two identical (=> uncertainties) mirrors perpendicularly rotated with respect to each other introduce more error sources, whose treatment is in principle possible with the techniques shown in this and the compagnion manuscript, but we cannot elaborate all possibilities in one paper.

**Comment**: The figure caption for figure 1 is anemic; identify each component; if I understood correctly, R should be identify as a lambda/2 waveplate in the text and in the figure caption

**Answer**: The caption has been improved following the suggestion. To avoid misunderstandings, an explicit comment is performed in the caption (caption Figure 1):

Figure 1: Lidar scheme based on functional blocks (adapted from Freudenthaler, 2016a). From right to left, laser ($I_L$), steering optics ($M_E$), atmosphere ($F$), receiving optics ($M_O$), calibrator ($C$), additional rotation of the PBS by 90° (**R**)), polarising beam-splitter cube (transmitted (T) and reflected (R) matrices, $M_T$ and $M_R$), detectors ($\eta_T$ and $\eta_R$), and the transmitted (T) and reflected (R) signals ($I_T$ and $I_R$).

and in the text (page 7, lines 14-18):

The parallel polarized component of the emitted laser beam can be detected either in the transmitted or in the reflected path behind the PBS. This depends on the orientation $\Psi$ of the PBS with respect to the laser polarization. We consider this by means of a rotator, $\mathbf{R}(\psi)$, (Eq. 2.14) before the PBS (see Fig. 1). For $\Psi = 90°$, the reflected and transmitted signals correspond to the parallel and perpendicular polarized components, respectively, and vice versa for $\Psi = 0°$.

**Comment**: In eq 2.1, why 2 alpha instead of alpha.

**Answer**: The Stokes' vector of the laser beam rotated by an angle alpha is calculated by

$$\mathbf{I}_L(\alpha) = \mathbf{R}(\alpha)\mathbf{I}_L(0°) = \begin{pmatrix} 1 & 0 & 0 & 0 \\ 0 & cos(2\alpha) & -sin(2\alpha) & 0 \\ 0 & sin(2\alpha) & cos(2\alpha) & 0 \\ 0 & 0 & 0 & 1 \end{pmatrix} I_L \begin{pmatrix} 1 \\ a \\ 0 \\ 0 \end{pmatrix}$$

and thus, the $2\alpha$ comes from the Müller matrix of a rotation by an angle $\alpha$.

**Comment:** In Eq. 2.13, why 2 beta instead of beta.

**Answer**: See previous comment.

**Comment**: For Eq. 2.2 a reference is required.

**Answer**: The references Lu and Chipman, 1996 and Chipman, 2009 have been included (page 3, line 28).

**Comment**: For Eq 2.23, it should be specify it is the scattering matrix for randomly oriented particle; reference to Michenko [2], Gimmestad [3] and Roy [6] should be made;

**Answer**: We assume the referee makes reference to the Eq. 2.27 (AMTD version).The references of van de Hulst, 1957 and Mishchenko and Hovenier, 1995 have been included (page 8, line 29).

**Comment**: Eq. 2.27, how Gs and Hs are obtained?

**Answer**: We assume the referee makes reference to the Eq. 2.31 (AMTD version). As stated in the manuscript, "the parameters $G_T$, $G_R$, $H_T$ and $H_R$, are determined solving the matrix multiplication of Equation 2.24 and separating the energy measured, $\mathbf{I}_S$, by the polarization parameter, $a$, as $I_S = G_S + aH_S$". We don't explain in detail this part because it is deeply done by Freudenthaler, 2016. Therefore, a comment pointing to the manuscript has

been included in this paragraph (page 9, line 19-21).

where the parameters $G_T$, $G_R$, $H_T$ and $H_R$, are determined solving the matrix multiplication of Equation 2.24 and separating the measured energy, $I_S$, in terms with and without the polarization parameter, $a$, as follow
(…)
(further details given by Freudenthaler, 2016a)

**Comment**: The authors should know that the average reader what to have a good idea of the meaning of a graphic by simply reading the caption. So for figure 3, 4, 5, 6, 7 and 8 spell out clearly all the meaning of variables. It is important.

**Answer**: Captions have been improved including detailed information of each variable (captions from Figure 3 to Figure 8).

---

## Author Comment (AC3) · 7 Jul 2016

Dear Dr. Miffre. We thank you very much for your comments, which give us the chance to elaborate some peculiarities of our approach for the numerical error calculation. This manuscript, published on 08 Feb 2016, uses the theoretical model as described in detail in the companion manuscript by Freudenthaler, 2016, doi:10.5194/amt-2015-338, which was published on 11 Feb 2016 as discussion paper together with this manuscript in the same journal and the same special issue, for the calculation of polarisation dependent systematic errors of a variety of lidar systems. We think that it is appropriate to split the long theoretical part and the part with the numerical error calculation in two papers in the same issue, so that one can refer to the other and not

all details and all references and related discussions have to be repeated again. However, the basic theory, which is necessary to follow the concepts of this manuscript, is included. This manuscript just uses the theory of amt-2015-338 for a comprehensive numerical error assessment with a complete search of the systematic, polarisation dependent error space. In this error assessment we include the error space of the calibration factor depending on the uncertainties of all model parameters. Such an error assessment, which includes simultaneously all the error sources of our model, seems not possible by means of analytical error analysis; at least nobody showed it yet. Furthermore, in the manuscript we compare the complete uncertainty of the volume linear depolarisation ratio of eight different lidar systems with various setups and calibration techniques on the basis of the proposed model in amt-2015-338. These results can be used as a reference in scientific works employing the investigated lidar systems. We have the impression that the majority of your comments rather pertain to the theoretical part amt-2015-338. In the view, that the two manuscripts belong together, we will answer your comments in the attached supplement.

Please also note the supplement to this comment:
http://www.atmos-meas-tech-discuss.net/amt-2015-339/amt-2015-339-AC3-supplement.pdf

**Supplement:**

Reply to Interactive comment to amt-2015-339

**Dear Dr. Miffre.**

We thank you very much for your comments, which give us the chance to elaborate some peculiarities of our approach for the numerical error calculation.

This manuscript, published on 08 Feb 2016, uses the theoretical model as described in detail in the companion manuscript by Freudenthaler, 2016, doi:10.5194/amt-2015-338, which was published on 11 Feb 2016 as discussion paper together with this manuscript in the same journal and the same special issue, for the calculation of polarisation dependent systematic errors of a variety of lidar systems. We think that it is appropriate to split the long theoretical part and the part with the numerical error calculation in two papers in the same issue, so that one can refer to the other and not all details and all references and related discussions have to be repeated again. However, the basic theory, which is necessary to follow the concepts of this manuscript, is included.

This manuscript just uses the theory of amt-2015-338 for a comprehensive numerical error assessment with a complete search of the systematic, polarisation dependent error space. In this error assessment we include the error space of the calibration factor depending on the uncertainties of all model parameters. Such an error assessment, which includes simultaneously all the error sources of our model, seems not possible by means of analytical error analysis; at least nobody showed it yet.

Furthermore, in the manuscript we compare the complete uncertainty of the volume linear depolarisation ratio of eight different lidar systems with various setups and calibration techniques on the basis of the proposed model in amt-2015-338. These results can be used as a reference in scientific works employing the investigated lidar systems.

We have the impression that the majority of your comments rather pertain to the theoretical part amt-2015-338. In the view, that the two manuscripts belong together, we will answer your comments below.

1. In their manuscript, Bravo-Aranda et al. indicate that they propose a new tool to assess the lidar polarizing sensitivity and to quantify the error on the depolarization due to the unknown systematic errors. To my knowledge, such a discussion on the sensitivity and accuracy of polarization lidar measurements was proposed in (G. David et al., Appl. Phys B, 108, 197-216, 2012) through laboratory and field experiments, by quantitatively discussing all possible systematic errors, including the following ones that are here analyzed: a small-unpolarized component in the emitter laser beam, an imperfect separation of polarization components, a misalignment between the transmitter and receiver polarization axes. Could the authors situate their work in regards to other published works and identify the novelty of their work? I feel quite surprised that this paper is not quoted.

Answer: As it is stated in the abstract, 'this work presents a new tool to assess the lidar polarizing sensitivity and to estimate the systematic error of the volume linear depolarization ratio ( $\delta$ ), combining the Stokes-Müller formalism and the Monte Carlo technique'. The mentioned citation has been included in the manuscript (page 2, line 17). David et al., 2012, is referenced in amt-2015-338. It discusses the influence of several error sources individually, not in combination, and not including the rotational errors between them or the retardation of the beamsplitters. Furthermore, the neglects are special for the instrument described there.

2. As a reader, it seems to me that the term of "polarizing sensitivity", as proposed all along the manuscript, needs some clarification. Could the authors add precision?

Answer: this phrase has been removed or replaced all along the manuscript.

3. The Stokes-Mueller matrix formalism is here used by introducing the Mueller matrix of the emitter optics and of the receiver optics with reference to Freudenthaler (2016a) all along the manuscript, a reference that is not yet published and hence not yet reviewed. For the sake of clarity, I propose to the authors to use reference to the publication by G. David et al. (Opt. Exp., 21, 16, 18624-18639, 2013) where with my co-authors, we stated the Stokes-Mueller matrix formalism for a pulsed laser source in the lidar backscattering geometry by introducing the same emitting and receiver optics Mueller matrices and the corresponding formalism. Could the authors situate their work in regards to this published work? I feel quite surprised that this paper be not quoted.

Answer: Please see the general comment above.

Furthermore, there was a mistake in the References of the manuscript using a title of Freudenthaler (2016a) which had been changed in the meantime. We apologize for that.

David et al., 2013, is referenced in amt-2015-338. As explained above, we believe that Freudenthaler (2016a) (i.e. amt-2015-338) can be used as a reference. In fact, it has to be used, because David et al., 2013, or any other paper do not provide the complete model presented there. Furthermore, David et al., 2013, are not the first to propose the use of the Müller-Stokes formalism. However, Freudenthaler (2016a) cannot be replace by the other one because the simulator is based on the new general formula presented inside.

4. When reading the manuscript, the reader gets the impression that the approach that is here proposed (the polarization lidar simulator) is the only possible one to address the sensitivity and the accuracy of lidar depolarization measurements. In David et al. (APB, 2012), we used a somewhat different approach by addressing the detector transfer matrix, which allows, as being diagonal, to perform a robust calibration, after minimizing all possible effects on the emitter optics. What additional information(s) do the authors get by introducing the Stokes-Mueller matrix formalism compared to our contribution? Whether this formalism is necessary or not should be clearly stated for potential readers.

**Answer**: As described above, we present a complete numerical search of the error space including all parameters of the model. We believe it is sufficiently clear for the reader that this approach excels analytical error calculations which consider only a few error sources individually.

5. The manuscript only deals with 1\_-polarization lidars while many papers have been published with 2\_polarization lidars such as Shimizu et al., 2004 or Tesche et al., 2009, as well-known. Such polarization lidar stations rely on wavelength-dependent components, that may attenuate the backscattering intensity and also modify the polarization state of the backscattered radiation in agreement with Fresnel's formulas. How do the authors account for this important contribution? Is their conclusion similar to that published in David et al. (APB, 2012)? Different?

**Answer**: In our manuscripts we analyse the possible errors of each individual signal channel of the lidar model. Each channel is only used at one wavelength. The channels used at other wavelengths have to be described with

the optical parameters for that wavelength. We do not deal with errors arising from a combined analysis of signals at different wavelengths.

6. Using a polarization simulator may be interesting but it is not to my knowledge the only possible approach. To address accurate lidar depolarization measurements or to calibrate the lidar depolarization, laboratory experiments at exact 180\_ are nowadays available and can be used to quantitatively address this quantity, even for non-spherical particles such as mineral dust particles, and at two-wavelengths, as we recently published (Miffre et al., JQSRT 2016). I think adding such information may be useful for potential readers of your manuscript, as it complements your approach. Indeed, a robust calibration has to rely on accurate laboratory measurements. That's why, I am proposing to add this reference that complements your work.

**Answer**: This manuscript is about error calculation, not about calibration. In the theoretical part, which describes several polarisation calibration techniques, we reference some publications describing alternative calibration techniques. However, Miffre et al., JQSRT 2016, do not describe a new technique but use the well-known method of Alvarez et al., J. Atmos. Ocean. Technol. 2006. Furthermore, a single laboratory calibration would not be sufficient. The actual calibration factor of a lidar has to be determined repeatedly over time in order to detect or exclude temporal changes.